# An adaptive stress response that confers cellular resilience to decreased ubiquitination

Liam C. Hunt [1,5,6], Vishwajeeth Pagala[2,6], Anna Stephan[1], Boer Xie[2], Kiran Kodali[2], Kanisha Kavdia[2], Yong-Dong Wang [3], Abbas Shirinifard[1], Michelle Curley [1], Flavia A. Graca[1], Yingxue Fu [2], Suresh Poudel[2], Yuxin Li[2], Xusheng Wang [2], Haiyan Tan[2], Junmin Peng [1,2,4] & Fabio Demontis [1] ✉

Ubiquitination is a post-translational modification initiated by the E1 enzyme UBA1, which transfers ubiquitin to ~35 E2 ubiquitin-conjugating enzymes. While UBA1 loss is cell lethal, it remains unknown how partial reduction in UBA1 activity is endured. Here, we utilize deep-coverage mass spectrometry to define the E1-E2 interactome and to determine the proteins that are modulated by knockdown of UBA1 and of each E2 in human cells. These analyses define the UBA1/E2-sensitive proteome and the E2 specificity in protein modulation. Interestingly, profound adaptations in peroxisomes and other organelles are triggered by decreased ubiquitination. While the cargo receptor PEX5 depends on its mono-ubiquitination for binding to peroxisomal proteins and importing them into peroxisomes, we find that UBA1/E2 knockdown induces the compensatory upregulation of other PEX proteins necessary for PEX5 docking to the peroxisomal membrane. Altogether, this study defines a homeostatic mechanism that sustains peroxisomal protein import in cells with decreased ubiquitination capacity.

Protein ubiquitination is an important post-translational modification that regulates protein localization, function, and degradation[1–8]. Protein substrates are ubiquitinated through the concerted actions of two E1 ubiquitin-activating enzymes (the predominant UBA1, previously known as UBE1, and to a lower extent the non-canonical UBA6), ~35 E2 ubiquitin-conjugating enzymes, and ~620 E3 ubiquitin ligases in humans[9–13]. This enzymatic cascade starts with an E1 that binds to and transfers a ubiquitin or ubiquitin-like molecule to an E2 that subsequently recruits a client E3, which in turn acts as an enzyme or as a scaffold to link ubiquitin typically to a lysine (but also to other residues) on a specific target protein[9–13]. Mono- and poly-ubiquitination constitute a complex code that regulates protein localization,

function, and degradation via the proteasome and the autophagy-lysosome system[1,2,4,5,14–23]. E2s have the center stage in the ubiquitination cascade by determining which E3s are recruited as well as the topology and length of ubiquitin chains[1,24–26]. However, the specificity of E2s in determining protein abundance remains largely uncharted.

UBA1 is ubiquitously expressed and is the primary E1 responsible for initiating E2-mediated ubiquitination in all tissues[27]. Consequently, overt loss of UBA1 is cell lethal[28,29]. However, partial reduction in UBA1 activity is surprisingly well tolerated and seemingly results in the derangement of the function of only a few cell types. Specifically, UBA1 hypomorphic mutations cause VEXAS syndrome, an adult-onset systemic inflammatory condition[30,31] that leads to progressive bone

[1]Department of Developmental Neurobiology, St. Jude Children's Research Hospital, 262 Danny Thomas Place, Memphis, TN 38105, USA. [2]Center for Proteomics and Metabolomics, St. Jude Children's Research Hospital, Memphis, TN 38105, USA. [3]Department of Cell and Molecular Biology, St. Jude Children's Research Hospital, Memphis, TN 38105, USA. [4]Department of Structural Biology, St. Jude Children's Research Hospital, Memphis, TN 38105, USA. [5]Present address: Department of Biology, Rhodes College, 2000 North Pkwy, Memphis, TN 38112, USA. [6]These authors contributed equally: Liam C. Hunt, Vishwajeeth Pagala. ✉e-mail: Fabio.Demontis@stjude.org

marrow failure because of defects in the function of hematopoietic stem cells[32–34]. Moreover, missense mutations that reduce UBA1 expression cause infantile spinal muscular atrophy X-linked 2 (SMAX2), a disease characterized by muscle weakness[35–40]. Altogether, although decreased UBA1 function may also contribute to the etiology of other diseases[41–46], partial reduction in the levels of UBA1 and in the activity of E2 ubiquitin-conjugating enzymes appears to be relatively well tolerated by many cell types, suggesting the presence of homeostatic stress responses that confer resilience to partial defects in UBA1 and E2 function.

Despite the identification of UBA1 mutations and reduced function as the culprit for VEXAS and SMAX2, it remains largely unexplored how partial reduction in UBA1 activity impacts the proteome. While it is well-established that disease-associated UBA1 mutations reduce E2-mediated ubiquitin conjugation[30,31], it is unknown whether the proteome is generally affected by a moderate decline in UBA1/E2 activity or whether, alternatively, the turnover of certain proteins is particularly sensitive to suboptimal UBA1 function. Therefore, an unresolved question is whether a subset of the proteome is primarily affected by a moderate reduction in ubiquitin conjugation and whether such changes in turn impact cellular function.

Here, we utilized deep-coverage TMT mass spectrometry in human HEK293T cells to identify the proteome subsets that are most impacted by a global reduction in E1 function, which was induced by UBA1 RNAi and by combining siRNAs to target 32 E2s expressed in HEK293 cells ("E2 combo"). By utilizing RNA-seq, we have further defined the proteomic changes that arise from UBA1/E2combo RNAi independently from changes in mRNA levels. Moreover, we have utilized a similar strategy to determine the proteomic changes caused by the knockdown of each E2. Further large-scale analyses identify E2-specific biases in linkage-specific ubiquitination, and the analysis of the E2 interactome indicates that similarity in target protein modulation is only in part explained by E2 cross-interactions. Altogether, these analyses have uncovered the proteome subsets that are modulated by global and individual E2 knockdown and have therefore defined what part of the proteome is affected by a moderate, partial reduction in ubiquitin conjugation.

In addition to charting how the proteome is affected by the knockdown of UBA1/E2s, we have also identified key cellular adaptations that occur in response to general and targeted reduction in ubiquitin conjugation. In particular, we have found that global reduction in UBA1/E2 function increases the import of peroxisomal proteins from the cytosol into the peroxisomal matrix. Previous studies have demonstrated that peroxisomal protein import relies on the ubiquitin-dependent peroxisome-to-cytosol cycling of the cargo receptor PEX5[47–50]. On this basis, a predicted detrimental outcome of decreased ubiquitination would be the derangement of peroxisomal protein import. However, we find that partial UBA1/E2 knockdown paradoxically promotes peroxisomal protein import via the counterbalancing upregulation of other PEX proteins (peroxins) different from PEX5 that are necessary for the docking to the peroxisomal membrane of the cargo receptor PEX5. Moreover, we find that the increase in peroxisomal protein import observed upon global reduction in UBA1/E2 function can be recapitulated by RNAi for the ubiquitin-conjugating enzyme UBE2D in human cells and in *Drosophila* skeletal muscle. Mechanistically, like UBA1 and E2combo RNAi, UBE2D knockdown reduces the turnover and hence increases the levels of peroxins necessary for docking PEX5 and mediating the import of peroxisomal proteins from the cytosol into the peroxisomal matrix.

In summary, this study defines how the proteome is affected by a moderate, partial reduction in ubiquitin conjugation and the consequent organelle adaptations that occur in response to UBA1/E2 decline. This study, therefore, highlights proteomic changes and cellular adaptations that are corollary to UBA1/E2 decline and that contribute to maintaining homeostasis in cells with partial defects in ubiquitin conjugation.

## Results

### Integrated proteome and transcriptome analyses identify protein subsets that are modulated by E2 ubiquitin-conjugating enzymes in a transcription-independent manner

E2 ubiquitin-conjugating enzymes are widely expressed across human tissues and in human HEK293T cells (Supplementary Fig. 1a). On this basis, we have utilized HEK293T cells to test the specificity of E2s in regulating protein abundance. To this purpose, cells were transiently transfected with siRNAs targeting individual E2s or groups of related E2s with high sequence homology (e.g. UBE2D1/2/3), which were targeted together by combining E2-specific siRNAs to avoid genetic redundancy. After 3 days from transfection, cells were harvested and analyzed by ultra-deep whole proteome profiling via TMT (tandem mass tag) mass spectrometry and by RNA-seq, compared to control cells treated with NT siRNAs. Examination of the E2 mRNA and protein levels revealed that, as expected, siRNA-targeted E2s were downregulated specifically, i.e. without significantly impacting the levels of other E2s not targeted by the siRNAs (Fig. 1a, b). Moreover, there was no significant impact of E2 RNAi on the activity of the proteasome, indicating that protein changes induced by E2 RNAi do not arise from general changes in proteasome-mediated proteolysis (Supplementary Fig. 1b).

Analysis of the TMT mass spectrometry and RNA-seq data resulting from the knockdown of individual and related E2s indicates that, compared to NT control siRNAs, several proteins are upregulated and downregulated without any corresponding mRNA changes, i.e. independently from transcriptional responses induced by E2 siRNAs. On average, each TMT quantified 10700 proteins: 5132 of these (mapping to 4676 DAVID IDs) were modulated by one or more E2s (Fig. 1c, Supplementary Fig. 2, and Supplementary Data 1). On average, the knockdown of some E2s leads primarily to protein downregulation (e.g. UBE2L6, UBE2M, UBE2N, UBE2S) whereas knockdown of other E2s is biased towards protein upregulation (e.g. UBE2D1/2/3, UBE2F, UBE2G1, UBE2G2).

E2 knockdown may remodel the proteome by impacting E2-mediated mono- and poly-ubiquitination, which regulates protein import, localization, and degradation[1,2,4,17–20]. To determine the impact of E2 knockdown on ubiquitination, we have utilized JUMPptm, an integrative computational pipeline for exploring post-translational modifications in TMT proteomics datasets[51]. Overall, these JUMPptm analyses identified the ubiquitination status of the most abundant proteins detected by TMT and highlighted E2 biases in linkage-specific ubiquitination (Fig. 1d). In particular, because of its consistent detection across the conditions analyzed, we examined the ubiquitination status of RPS27A, a fusion protein consisting of ubiquitin and the ribosomal protein S27A, which is proteolytically processed to generate ubiquitin. Knockdown of 13 out of 24 individual or related E2s reduced K48-linked ubiquitination whereas this increased upon heat shock (Fig. 1d). Other linkage-specific modifications (K6, K11, K27, K29, K33, and K63) were affected by fewer E2s. Specifically, K6-linked ubiquitination was significantly reduced by knockdown of UBE2A/B, UBE2D1/2/3, UBE2E1/2/3, and UBE2T whereas K11-linked ubiquitination was significantly reduced by UBE2D1/2/3. UBE2V1/2 knockdown increased K27 ubiquitination, an effect that is presumably due to the compensatory activity of other non-targeted E2s (Fig. 1d, Supplementary Fig. 3, and Supplementary Data 2).

Analysis of the proteins that are regulated by E2 RNAi without significant mRNA changes indicates that many proteins are modulated by single E2s (2355; gray) and by 2 to 4 E2s (2162; green) whereas only 159 proteins (pink) are cross-regulated by 5 to 9 E2s (Fig. 1e). GO term analyses revealed that several categories are generally enriched among E2-regulated proteins, including Ubl conjugation, endoplasmic reticulum, peroxisome, mitochondrion, and DNA repair (Fig. 1f–h). Conversely, other categories are enriched only in specific subsets: for example, cellular senescence is a category that is over-represented

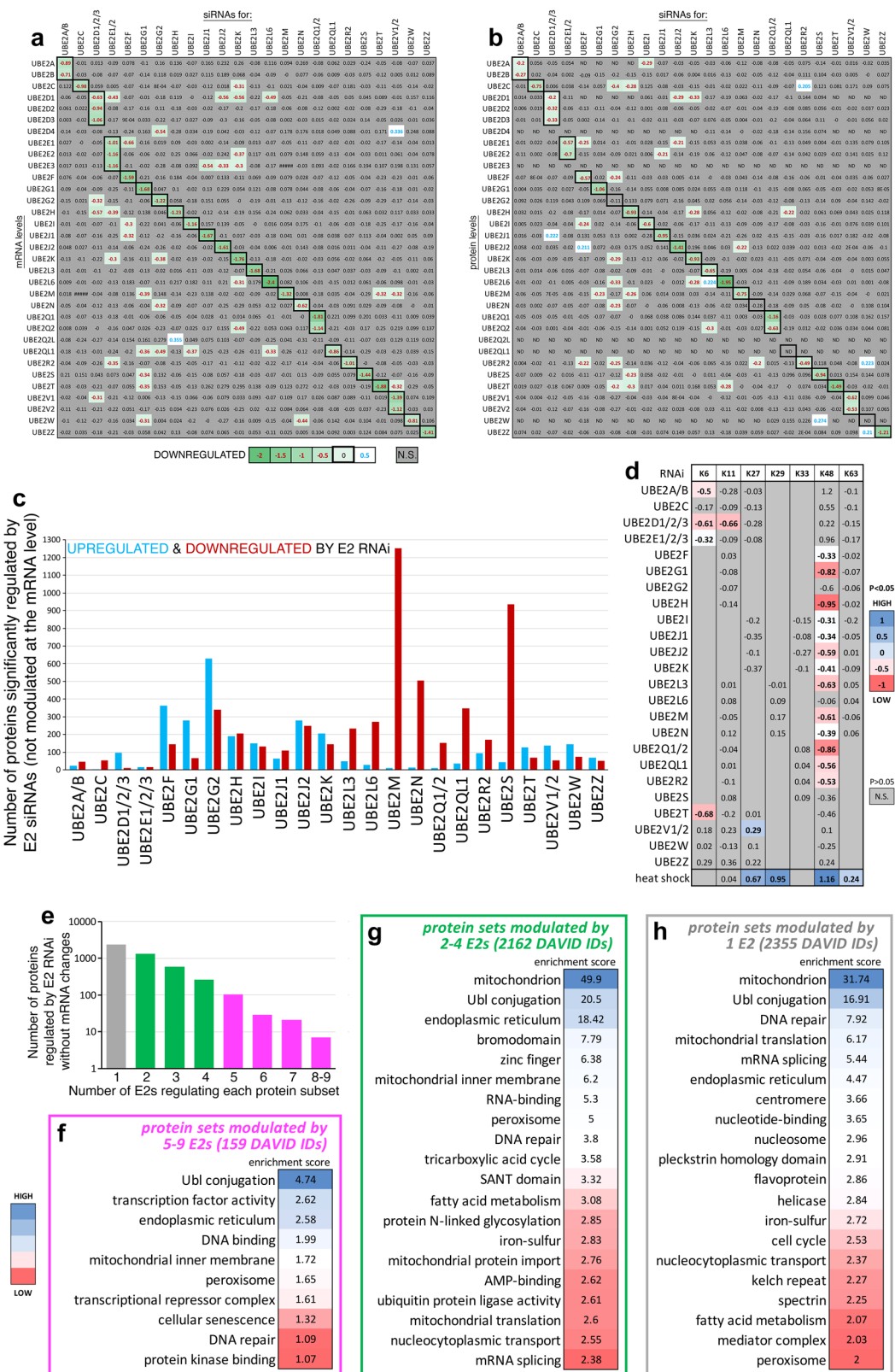

among the proteins regulated by 5-9 E2s whereas bromodomain and centromeres are categories that are enriched among the proteins modulated by 2–4 and by single E2s, respectively (Fig. 1f–h). Altogether, these analyses identify protein categories that are enriched among E2-regulated proteins.

We then further analyzed E2-regulated proteins to determine whether they are enriched for short-lived and/or ubiquitinated proteins (Supplementary Fig. 4a, b). To this purpose, we cross-compared our deep-coverage TMT data (Fig. 1) with the datasets of short-lived and ubiquitinated proteins that have been previously generated in HEK293T cells[52,53]. These analyses revealed that there is overall no enrichment for short-lived proteins among E2-regulated proteins. However, proteins upregulated by RNAi for UBE2E1/2/3, UBE2M, UBE2N, and UBE2S display lifetimes that trend towards lower

**Fig. 1 | Proteomic changes induced by knockdown of E2 ubiquitin-conjugating enzymes in human cells.** E2 knockdown in HEK293T cells was obtained via siRNAs for individual E2s or for groups of related E2s with high sequence homology (e.g. UBE2D1/2/3), which were targeted together by pooling E2-specific siRNAs. E2 siRNAs reduce (green) the mRNA (**a**) and protein (**b**) levels of the targeted E2s (boxed) specifically, i.e. without affecting other E2s, compared to non-targeting (NT) siRNAs. Significant ($P < 0.05$; unpaired two-tailed $t$ test) transcriptional changes are highlighted in bold red (downregulated) and blue fonts (upregulated). The extent of downregulation is displayed in green shades whereas non-significant changes are shown in gray. **c** Deep-coverage TMT mass spectrometry identifies the proteome subsets that are regulated by E2 RNAi. Compared to NT siRNAs, E2 RNAi leads to significant protein upregulation (blue) and downregulation (red) ($P < 0.05$; Log$_2$FC > 0.2 and <−0.2) that do not arise from corresponding changes in mRNA levels, defined by RNA-seq. This data represents 6 sets of 16-plex TMTs of E2 siRNAs

($n = 3$ biological replicates/group), each with its own control NT siRNAs ($n = 4$ biological replicates). On average, each TMT set quantified 10700 proteins: 5132 of these (mapping to 4676 DAVID IDs) were modulated by ≥1 E2s. **d** JUMPptm analyses identify linkage-specific ubiquitin modifications modulated by E2s. In particular, the knockdown of many E2s reduces K48-linked ubiquitination whereas this increases upon heat shock. Other linkage-specific modifications (K6, K11, K27, K29, K33, and K63) are affected by fewer E2s. Significant downregulation ($P < 0.05$) of these modifications by E2 RNAi is displayed in bold fonts and shades of red whereas non-significant changes ($P > 0.05$) are shown in gray. Supplementary Fig. 3 reports the precise $P$ values (one-way ANOVA). **e** Proteins that are regulated by E2 RNAi without significant mRNA changes are reported on the $y$-axis whereas the number of E2s regulating each protein subset is reported on the $x$-axis. Several GO categories are enriched among protein sets that are modulated by 5–9 E2s (**f**), 2–4 E2s (**g**), and by single E2s (**h**). Source data are provided in the Source data file.

values (Supplementary Fig. 4b). Moreover, ~64% of the proteins that we found to be regulated by E2s were previously reported to be ubiquitinated in HEK293T cells (Supplementary Fig. 4c).

## Interactome mapping of E2 ubiquitin-conjugating enzymes reveals E2 cross-interactions, E2-E3 pairs, and E2 association with cellular complexes in human cells

The interaction between E2 ubiquitin-conjugating enzymes and E3 ubiquitin ligases as well as other proteins necessary for ubiquitin chain editing has largely been explored on an individual basis for only some E2s. Previous studies have unraveled E2-E3 interactions primarily via yeast two-hybrid[54,55], a technique that has high rates of false positives and false negatives and that generates purely qualitative interaction data in a non-native environment[56]. Altogether, there is an incomplete understanding of the network of interactions that govern the function of human E2s.

We have previously found that the C-terminally FLAG-tagged E2 enzyme UBE2B readily co-purifies with associated E3s including RNF20, RNF40, UBR1, UBR2, and UBR4, and that the UBE2B-UBR4 interaction is physiologically relevant in vivo, where it regulates muscle cell growth and proteostasis in *Drosophila* and mice[57,58]. On this basis, we have C-terminally FLAG-tagged 8 E1s (including UBA1) and all 28 human E2s expressed by human HEK293T cells, transfected these plasmids, and immunoprecipitated each E1/E2-FLAG with anti-FLAG antibodies. Subsequently, the interacting proteins were identified by spectral counting via mass spectrometry by assaying 3 biological replicates for each E1/E2-FLAG bait and 10 controls (i.e. no-FLAG baits).

An example of E1/E2 affinity purification with Coomassie staining, western blotting, and mass spectrometry is reported in Fig. 2a, b, which shows enrichment of the UBE2A bait as well as of E3 preys. The spectral counts were compared by Significance Analysis of INTeractome (SAINT)[59], which determined significantly enriched interactors (SAINT ≥ 0.65) compared to control samples: this analysis yielded 1171 significantly interacting pairs for 28 unique E2 baits, which were all significantly enriched in their own samples, together with 515 unique preys. In addition to SAINT, we also compared detected proteins with the Contaminant Repository for Affinity Purification (CRAPome)[60] and additionally compared maximum spectral counts for proteins detected in association with E2 baits, which were consistently higher than those retrieved from control (no bait) purifications. Altogether, these analyses indicate that we have identified reliable E2 interactors, which include E3 ubiquitin ligases and enzymes with deubiquitinase activity (DUBs) or accessory to ubiquitin chain editing complexes (Fig. 2c), as estimated based on previous annotations[61]. Specifically, this interactome demonstrates a preference of E2s for certain E3s, as exemplified by UBE2D1/2/3 that preferentially interact with HECT domain-containing E3 ligases (HECTD1, HERC1, and HUWE1).

Very few E2 baits returned no E3/DUB partners: these were UBE2L3, UBE2M, UBE2Q2 and UBE2U. A possible explanation may consist in the transient nature of E2-E3 interactions and in the fact that

some E3s are expressed below the detection threshold in HEK293 cells (Supplementary Fig. 5a−d). However, interestingly, these E2s physically interacted with other E2s, which could indicate they act predominantly as an accessory to other E2-E3 ubiquitin chain editing complexes without binding to an E3 directly. Importantly, these prevalent cross-interactions between E2s (Fig. 2d) are largely independent of amino acid sequence homology between E2s (Fig. 2e). Altogether, this data suggests that E2s cooperate not just with E1s and E3s to conjugate ubiquitin-like proteins to targets but also with other E2s, and that these interactions are particularly common for certain E2s such as UBE2J1, UBE2N, and UBE2S.

Beyond E2 interactors, this IP-MS interactome mapping also confirms known interactions of E2s with ubiquitin-related proteins (Fig. 2f, g, Supplementary Fig. 5, and Supplementary Data 3-4). For example, UBE2I interacts with SUMO1-3, UBA2, and SAE1, consistent with UBE2I being the major E2 for conjugating the ubiquitin-like modifier SUMO via the E1 heterodimer composed by UBA2/SAE1[62,63]. In contrast to UBE2I being the only E2 for SUMOylation, UBE2F, and UBE2M were both found to interact with UBA3/NAE1 for conjugating the ubiquitin-like modifier NEDD8[64,65]. UBA1, the major E1 ubiquitin-activating enzyme, was found to interact with a number of E2s including UBE2A, UBE2B, UBE2D1/2/3, UBE2E1/2, UBE2G1/2, UBE2J1/2, UBE2L6, UBE2N, UBE2R1 (CDC34), and UBE2S. Interestingly, UBE2L6 also interacted with UBA6, which in addition to conjugating ubiquitin can conjugate the ubiquitin-like protein FAT10[66]. Although several E2s did not affinity purify with E1s, it may be possible that they interact less stably with an E1 and/or that, as indicated by the prevalence of E2 cross-interactions, they may act as ubiquitin-conjugating enzymes through interactions with other E2s that associate with an E1. For example, UBE2V1/2 (which did not interact with an E1) was found in a complex with UBE2N, which in turn interacted with UBA1 (Fig. 2f, g). Although this would suggest that UBE2V1/2 should co-immunoprecipitate with a UBA1-UBE2N complex, this did not occur, presumably because these interactions are dynamic or mutually exclusive. Moreover, if the interaction of UBE2V1/2 to UBA1 is indirect and occurs via UBE2N, only a small fraction of UBE2V1/2 may associate with the UBA1 bait and thus be missed because below the MS detection threshold.

In addition to revealing E2 pairings with E1s, E3s, and DUBs, the affinity purification of E2 interactors also demonstrates several broad features of the E2 network, including overlap between the ubiquitin-proteasome (UPS) and autophagy-lysosome systems, and that some E2s (such as UBE2B and UBE2C) physically interact with proteasomal components (Fig. 2f–g). Altogether, interactome mapping in human cells indicates broad features of the E1/E2 network and predicts E2-E3 (and DUB) pairings that form ubiquitin chain editing complexes.

## E2 interactome similarity is in part predictive of the induction of analogous protein changes by E2 knockdown

Having established the network of physical interactions of E2 ubiquitin-conjugating enzymes, we next examined whether the E2

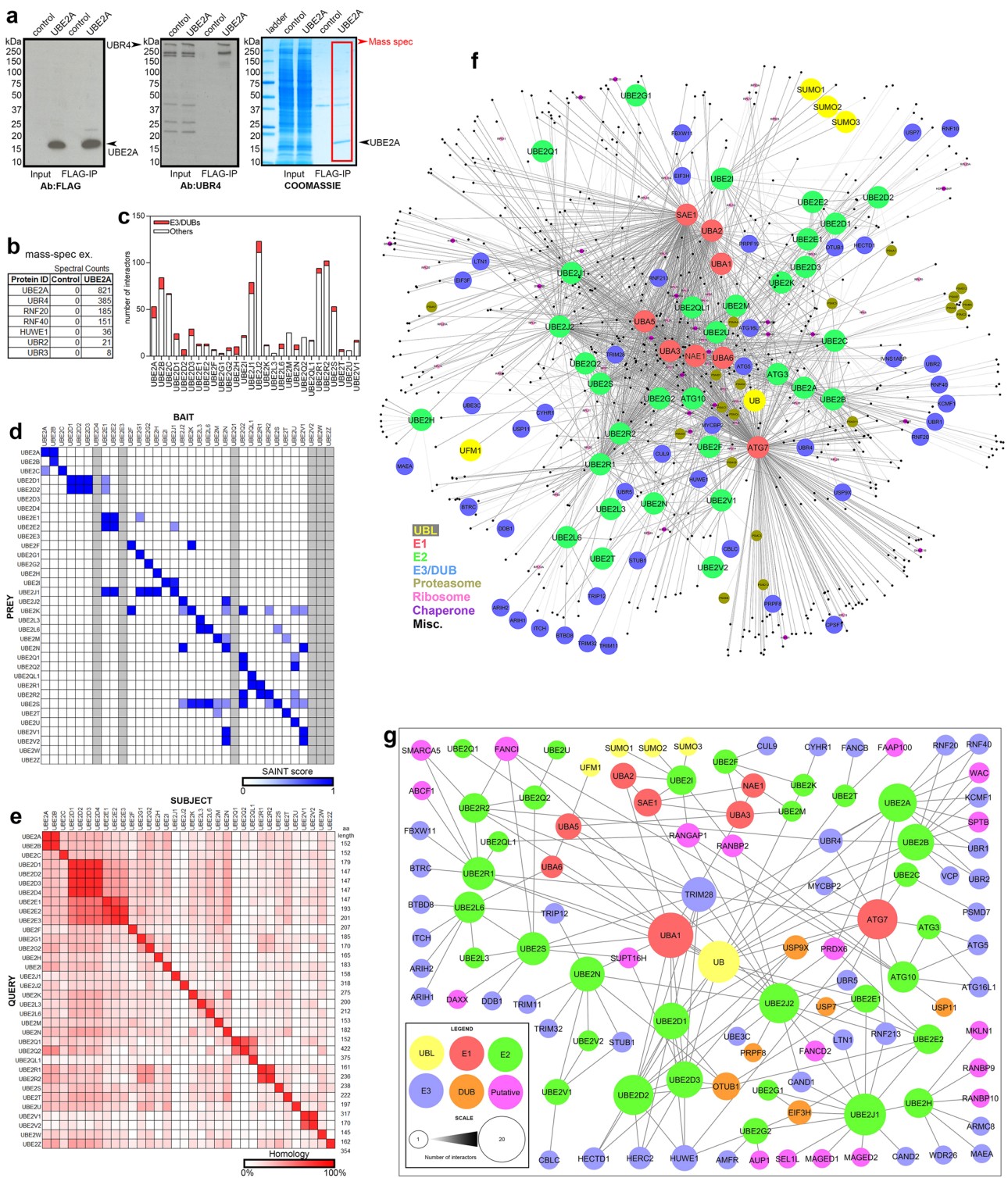

interactome correlates in any way with the changes in protein abundance identified from the deep-coverage mass spectrometry (Fig. 1). To this purpose, we cross-compared the similarity in the interactome of each E2 pair with the similarity in the protein changes induced by RNAi for each of these 2 E2 enzymes. Specifically, for each E2x versus E2y cross-comparison, the degree of similarity in the physical interactome was defined by the z-score of $R^2$ values obtained from the SAINT scores of E2x versus E2y. Likewise, the z-score of $R^2$ values obtained from the cross-comparison of TMT datasets from E2x RNAi versus E2y RNAi was utilized to define the similarity in the protein changes induced by the knockdown of these E2s. There was overall

little correlation between the interactome and the protein changes induced by most E2 pairs (Supplementary Fig. 6a) with some notable exceptions (Supplementary Fig. 6b). In particular, there was a similarity in the interactome and protein changes induced by some E2 pairs, i.e. UBE2J1-UBE2G2, UBE2K-UBE2F, UBE2T-UBE2F, UBE2J2-UBE2N, UBE2L3-UBE2T, UBE2M-UBE2F, and UBE2M-UBE2N. Such correlations were not due to the sequence homology of the E2s in the pair (Fig. 2e), apart from UBE2M-UBE2F (Supplementary Fig. 6c). Direct comparison of the percentage in E2 protein sequence identity to the similarity of protein changes induced by E2 RNAi revealed a substantial correlation only for the additional UBE2A/B-UBE2D1/2/3/4 pair

**Fig. 2 | Identification of E2 interactors in HEK293T cells via immunoprecipitation and mass spectrometry. a** An example of IP-MS of FLAG-tagged E2s to identify E2 interactors. Western blots for FLAG-tagged UBE2A detect the E2 bait, UBE2A, in input cellular extracts and following immunoprecipitation (IP) with anti-FLAG antibodies. Whole lanes of FLAG-IP for each E2 bait were excised and used for MS. **b** MS spectral counts were analyzed by SAINT to determine interacting proteins that were significantly enriched (SAINT score>0.65), which included E3s and deubiquitinating enzymes (DUBs). **c** Non-curated interactome of all proteins significantly interacting with the E1 and E2 baits (SAINT > 0.65); n = 3 biological replicates for each bait and n = 10 biological replicates for each control (no-FLAG baits). **d** Heatmap demonstrating that the FLAG-tagged E2 baits were detected and significantly enriched in their respective IP-MS. Cross-interactions were detected between E2s, even in cases where sequence homology is low (**e**). **f** Non-curated interactome of all proteins significantly interacting with E1 and E2 baits (SAINT > 0.65). The complex network integrates E1s for ubiquitin and ubiquitin-like proteins (UBLs), E2s, E3s, DUBs, proteasome and autophagy (ATGs) components, and other proteins. The UPS network of E1-E2-E3s interacts with the analogous E1-E2 autophagy network (ATG7, ATG3, and ATG10). Additionally, E2s such as UBE2C interact with proteasome components. **g** Curated interactome focused on UBLs, E1s, E2s, E3s and DUBs. This network, which is centered around ubiquitin (UB) and the UBA1 E1 enzyme, demonstrates specific UBL utilization, E1 and E2 cross-interactions, E2s with the same E3/DUB interactors and, conversely, with unique E3/DUB interactors. Source data are provided in the Source data file.

(Supplementary Fig. 6c): therefore, E2 sequence homology does not explain the similarity in the interactome and protein changes observed for the E2 pairs in Supplementary Fig. 6b. We next examined the subcellular localization of E2s by consulting the Human Protein Atlas (proteinatlas.org), which reports antibody-based immunostaining of endogenous E2s in HEK293T cells[67,68], and additional published datasets[69,70], which report the in silico prediction of E2 localization. In addition, immunostaining with anti-FLAG antibodies was utilized to determine the localization of FLAG-tagged E2s (Supplementary Fig. 7). These analyses indicate that E2 pairs that share a similarity in the interactome and in the proteome changes induced by their knockdown (Supplementary Fig. 6) also localize to the same subcellular compartment: UBE2J1-UBE2G2 in the endoplasmic reticulum; UBE2K-UBE2F, UBE2T-UBE2F, and UBE2M-UBE2F in the cytoplasm; and UBE2J2-UBE2N, UBE2L3-UBE2T, and UBE2M-UBE2N in the nucleus (Supplementary Fig. 7).

Altogether, these cross-comparisons indicate that the similarity in the E2 interactome is in some cases predictive of the occurrence of analogous changes in protein abundance upon E2 knockdown, and of similar E2 subcellular localization.

## Moderate global reduction in E2 activity reduces total and linkage-specific ubiquitination in human cells and impacts specific proteome subsets

We have profiled the proteomic changes induced by the knockdown of individual/related E2s (Fig. 1) and defined the E2 interactome (Fig. 2). By examining collectively the proteomic changes induced by RNAi for individual/related E2s, these integrated analyses have defined the E2-regulated proteome (Fig. 1). As an additional strategy to define the proteins that are most sensitive to partial E2 inhibition, we next utilized siRNAs for UBA1, the major E1 ubiquitin-activating enzyme necessary for E2-mediated ubiquitination[27], and the combination of siRNAs targeting distinct E2s to obtain the concomitant knockdown of multiple E2s ("E2 combo").

UBA1 knockdown primarily reduced the levels of detergent-insoluble ubiquitinated proteins compared to control cells treated with NT siRNAs (Fig. 3a). Additional analyses with antibodies specific for linkage-specific ubiquitination indicate that detergent-insoluble levels of proteins with K48 linkage-specific ubiquitination and SUMO1 modifications decline in cells treated with UBA1 siRNAs (Fig. 3b).

We next tested the combination of siRNAs targeting 32 E2s ("E2combo") and found that it leads to significant reduction in the mRNA levels of 19 E2s (Fig. 3c). Compared to treatment with the same amount of control non-targeting (NT) siRNAs, E2 combo RNAi led to partial decline (-20%) in ubiquitin conjugation (Fig. 3d, e), similar to what is found with UBA1 RNAi (Fig. 3a, b). Further analyses indicate that heat shock increases the levels of detergent-insoluble ubiquitinated proteins but that this is hindered by E2combo siRNAs, especially for total ubiquitination, K63 linkage-specific ubiquitination, and SUMO1 modifications (Fig. 3e).

Subsequently, we have profiled via RNA-seq and deep-coverage TMT mass-spectrometry the proteomic changes that are induced (independently from changes in mRNA levels) by UBA1 and E2combo siRNAs compared to control cells treated with NT siRNAs (Fig. 3f, h). There are several GO categories that are overrepresented among proteins that are significantly upregulated and downregulated (Fig. 3g, i) by siRNAs for UBA1 (Fig. 3f, g) and for E2combo (Fig. 3h–i) compared to control non-targeting (NT) siRNAs, and that are not significantly modulated at the mRNA level. Protein sets related to several organelles (e.g., Golgi, mitochondria, lysosomes) are modulated (Fig. 3g–i), indicating that organelle proteome and function might be regulated by a partial decline in ubiquitin conjugation. In particular, UBA1 RNAi and E2combo RNAi concordantly upregulate the levels of peroxisomal proteins (Fig. 3g–i). Altogether, these studies have identified a core set of proteins that are sensitive to partial, global knockdown of E2 function induced by UBA1/E2combo RNAi.

## Peroxins are upregulated in response to UBA1/E2combo RNAi and their turnover occurs at least in part via the proteasome

We have found that peroxisomal proteins are upregulated in response to RNAi for UBA1 and for E2combo (Fig. 3). On this basis, we analyzed the peroxisomal proteome (based on previous annotations[71,72]) to further dissect its modulation by UBA1/E2combo knockdown. These analyses indicate that UBA1 RNAi increases the levels of several peroxins (PEX3, PEX7, PEX11G, PEX13, PEX14, and PEX19), which regulate the biogenesis of peroxisomes[73–75] and/or peroxisomal protein import[47,49], and of other peroxisomal proteins, such as the transporter ACBD5[76] and the metabolic enzyme GNPAT (Fig. 4a)[77]. Likewise, E2combo RNAi increases the levels of peroxins (PEX3, PEX11B, PEX12, PEX14, PEX16) and of other peroxisomal proteins such as TMEM135 (Fig. 4b), which establishes peroxisome-lysosome membrane contacts[78,79]. Analysis of the protein and mRNA levels further confirms that PEX protein upregulation occurs without any significant changes in the corresponding PEX mRNA levels (Fig. 4c, d). Altogether, these analyses indicate that peroxins are particularly sensitive to moderate reduction in the function of UBA1 and E2s, as also found from the analysis of knockdown of individual/related E2s (Fig. 1e–h).

To further test this model, HEK293T cells were transfected with plasmids encoding for FLAG-tagged versions of some of the PEX proteins that are upregulated by UBA1/E2combo RNAi. In addition, also PEX5 was included in these analyses because of its key ubiquitin-dependent role in peroxisomal protein import[47–50], although its levels are not regulated by UBA1/E2combo knockdown (Fig. 4a, b). This experimental setup provides a system to monitor PEX protein modulation independently from possible changes in endogenous PEX gene expression (Fig. 4e). As expected, FLAG-tagged PEX proteins localize to functional peroxisomes identified by GFP-PTS1, a GFP protein with a peroxisomal targeting signal type 1[80] (Supplementary Fig. 8a). Subsequently, the levels of PEX-FLAG proteins were assessed by western blot: these analyses indicate that UBA1/E2combo RNAi increases the levels of exogenously-expressed PEX3-FLAG and PEX12-FLAG (Fig. 4e), confirming that modulation of PEX proteins occurs independently from transcription.

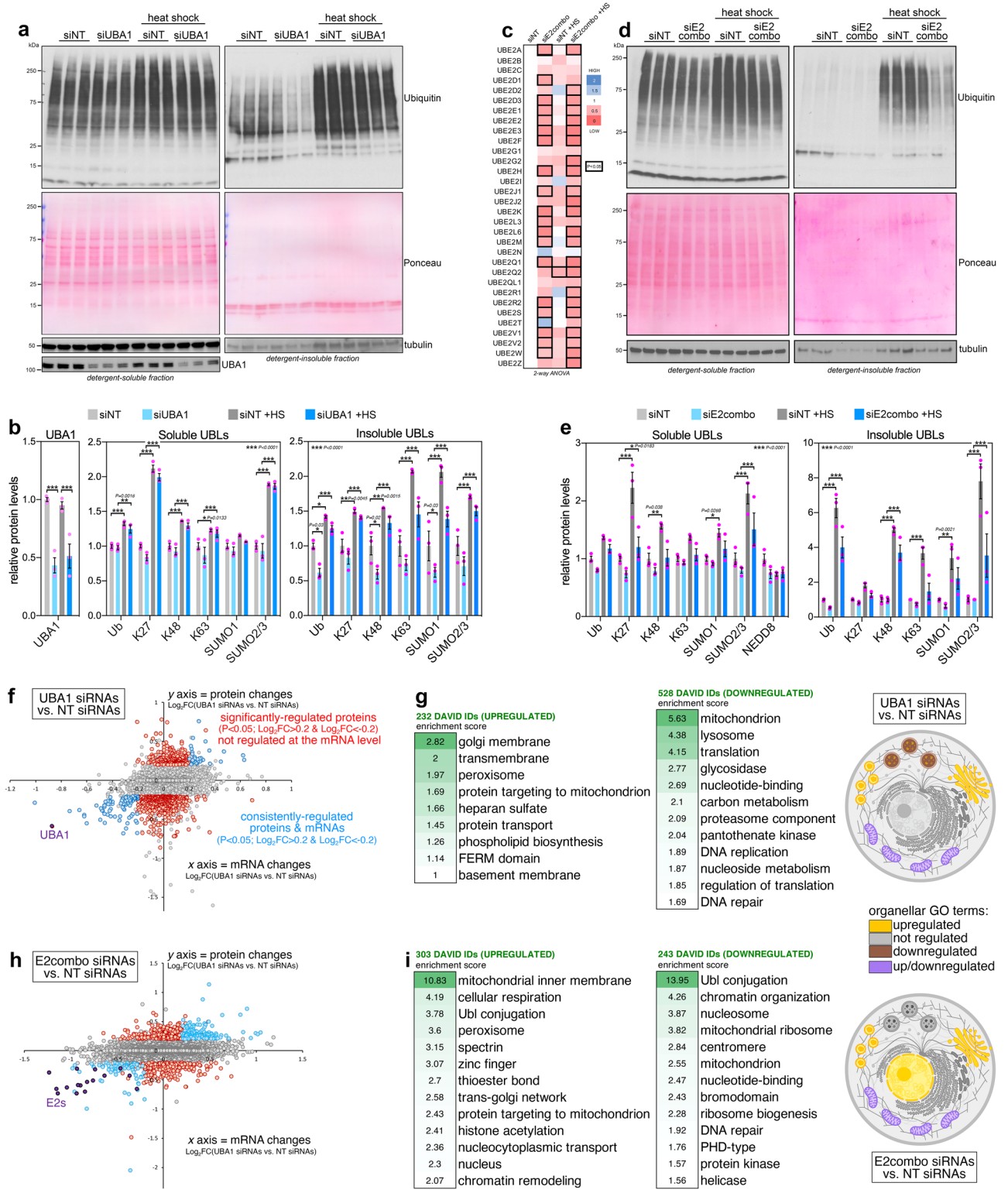

Based on these findings, we next examined whether peroxin turnover normally occurs via the ubiquitin-proteasome system. To this purpose, cells transfected with plasmids encoding for FLAG-tagged PEX proteins were treated with pharmacologic inhibitors of UBA1 (TAK243) and of the proteasome (MG132), either alone or in combination, and compared to mock-treated cells (DMSO). Immunoprecipitation of FLAG-tagged PEX proteins was followed by immunoblotting with anti-ubiquitin antibodies to determine the ubiquitination status of FLAG-tagged PEX proteins. Compared to controls, treatment with the proteasome inhibitor MG132 increased PEX ubiquitination and this was

largely prevented by concomitant treatment with the UBA1 inhibitor TAK243. Altogether, these findings indicate that peroxin turnover occurs at least in part via the ubiquitin-proteasome system (Fig. 4f and Supplementary Fig. 9).

## A moderate reduction in UBA1/E2 function promotes peroxisomal protein import in a PEX-dependent manner
Deep-coverage proteome profiling indicates that UBA1/E2combo RNAi upregulates the levels of PEX11, which promotes peroxisome biogenesis[73–75], and of other peroxins (e.g. PEX3/12/13; Fig. 4a, d) that

**Fig. 3 | Reducing E2 activity modulates several protein sets and organelle components.** A general, partial decline in E2 activity was obtained via UBA1 RNAi and via the concomitant RNAi-mediated knockdown of multiple E2s (siE2combo). **a** Western blot of HEK293T cells treated with siRNAs targeting UBA1 (siUBA1) versus control non-targeting siRNAs (siNT), with and without heat shock. **b** siUBA1 reduces the levels of detergent-insoluble ubiquitinated proteins. Heat shock (HS) increases the detergent-soluble levels of proteins modified with ubiquitin (total and K27, K48, and K63 linkage-specific ubiquitination) and with ubiquitin-like modifications (SUMO1 and SUMO2/3) but their levels are not modulated by siUBA1. The detergent-insoluble levels of ubiquitinated proteins (total and K48 linkage-specific ubiquitination, and SUMO1 modifications) decline in siUBA1- versus siNT-treated cells in the absence of heat shock; n = 3 biological replicates/group, mean ±SD, ***P < 0.001 (two-way ANOVA). **c** E2combo RNAi (a pool of siRNAs targeting 32 E2s) leads to a significant reduction in the mRNA levels of 19 E2s. The average of 3 biological replicates is shown, with red indicating a decline in E2 expression compared to control cells treated with the same amount of siNT. Bold-outlined boxes indicate a significant change (P < 0.05; unpaired two-tailed t test). **d, e** Detergent-soluble and insoluble ubiquitinated proteins decline in siE2combo-treated cells. Heat shock increases detergent-insoluble ubiquitinated proteins but this is hindered by E2combo siRNAs, especially for total ubiquitination and for K63 and SUMO1 modifications; n = 3 biological replicates/group, mean ±SD, ***P < 0.001 (two-way ANOVA). Proteomic and transcriptional changes induced by siUBA1 (**f**) and by siE2combo (**h**) versus siNT. Several proteins are upregulated and downregulated (P < 0.05; Log₂FC > 0.2 and <-0.2; red) without changes in the corresponding mRNAs. Proteins with corresponding mRNA changes induced by siUBA1 (P < 0.05; Log₂FC > 0.2 and <−0.2) are highlighted in blue and include siRNA-targeted UBA1 (**f**) and E2s (**h**), highlighted in violet. GO categories that are over-represented among proteins that are significantly (P < 0.05) upregulated (Log₂FC > 0.2) and downregulated (Log₂FC < −0.2) by siUBA1 (**g**) and by siE2combo (**i**) versus siNT, and that are not significantly modulated transcriptionally. Peroxisomal proteins are consistently upregulated in response to siUBA1 (**f**–**g**) and siE2combo (**h**, **i**). Source data are provided in the Source data file.

constitute receptor docking complexes that promote the import of peroxisomal proteins from the cytoplasm into the peroxisomal matrix[47–49]. Therefore, we hypothesized that PEX upregulation by UBA1/E2 RNAi may improve peroxisomal protein import and/or biogenesis.

To test this hypothesis, we have examined HEK293T cells transfected with GFP-PTS1, a GFP that is targeted to peroxisomes by the PTS1 peroxisomal targeting sequence[80]. These cells were also immunostained with antibodies for the peroxisomal membrane protein 70 (PMP70), which is an abundant transmembrane protein that is inserted into the peroxisomal membrane via chaperone-dependent mechanisms independent from PTS1-guided import[47–49,81,82]. Importantly, PMP70 has been found to mark both functional and dysfunctional (ghost) peroxisomes with defective PTS1-mediated protein import[83]. Conversely, GFP-PTS1 is absent from dysfunctional ghost peroxisomes, which are present in Zellweger syndrome and other disease states characterized by PEX mutation and/or peroxisomal dysfunction[84–86]. On this basis, we have utilized these markers to assess the extent of peroxisomal protein import in cells with UBA1/E2combo RNAi. To this purpose, the colocalization between PMP70 and GFP-PTS1 was estimated by using machine-learning computational analyses, and the relative proportion of functional (PMP70+ and PTS1+) and dysfunctional (PMP70+ and PTS1-) peroxisomes determined (Fig. 5a).

Because several peroxins that are upregulated by UBA1/E2 RNAi are components of the receptor docking complex that promotes PTS1-guided peroxisomal protein import, we hypothesized that UBA1/E2 RNAi increases the peroxisomal import of GFP-PTS1 and that this may depend on PEX. In agreement with this model, we found that UBA1/E2combo knockdown improves peroxisomal protein import, as indicated by the increase in the relative proportion of GFP-PTS1-positive versus GFP-PTS1-negative peroxisomes (Fig. 5b). On this basis, we next tested whether such improvement in peroxisomal protein import requires peroxins. To this purpose, cells were treated with E2combo siRNAs and either with control NT siRNAs or with a combination of siRNAs to knockdown PEX proteins (PEXcombo, i.e. PEX3, PEX11A, PEX11B, and PEX13 siRNAs) to counteract their upregulation in response to E2combo RNAi (Supplementary Fig. 8b). Importantly, the amount of NT and E2combo siRNAs was kept constant and analyzed in conjunction with equal amounts of either PEXcombo or control NT siRNAs to avoid titration effects. In agreement with the hypothesis, the proportion of functional peroxisomes (i.e., PMP70 puncta with GFP-PTS1) increased upon E2combo RNAi but such increase in peroxisomal protein import was blunted by PEXcombo knockdown (Fig. 5c). To further test this model, we employed siRNAs to individually target PEX3, PEX5, PEX11A + B, PEX12, and PEX13 (Fig. 5d). Consistent with Fig. 5b, c, we found that UBA1 and E2combo siRNAs increase peroxisomal protein import, as indicated by the higher proportion of functional

peroxisomes (i.e., PMP70 puncta with GFP-PTS1). Apart from PEX13 RNAi, siRNAs for other PEXs did not reduce peroxisomal import in control conditions (NT siRNAs). However, RNAi for PEX3, PEX5, PEX11A + B, PEX12, and PEX13 impeded (to different extents) the upregulation of peroxisomal protein import due to UBA1/E2combo knockdown (Fig. 5d). Altogether, these findings indicate that moderate reduction in UBA1/E2 function improves peroxisomal protein import and that this cellular adaptation is prevented by peroxin knockdown (Fig. 5a–d).

Some of the peroxins modulated by UBA1/E2combo RNAi (Fig. 4) have been primarily implicated in peroxisome biogenesis and proliferation, such as PEX11[47–49], and therefore they may modulate the number of peroxisomes rather than peroxisomal protein import. On this basis, we monitored the average number of peroxisomes (PMP70+ puncta) per cell but found no significant difference in the number of peroxisomes when comparing siRNAs for UBA1 and E2combo to control NT siRNAs (Supplementary Fig. 10).

## Partial knockdown of UBE2Ds induces a compensatory PEX upregulation that sustains peroxisomal protein import in human cells

We have found that global reduction in E2-dependent ubiquitination (UBA1 RNAi) and E2 function (E2combo RNAi) increases the levels of several peroxins and that this promotes peroxisomal protein import (Figs. 4, 5). We next surveyed the TMT data that have profiled the proteomic changes induced by individual/related E2s (Fig. 1) to determine whether certain E2s have a predominant role in PEX protein regulation. On this basis, we examined the significant changes in PEX protein levels (arising independently from mRNA changes) induced by RNAi for individual/related E2s (Fig. 1) and represented them cumulatively for each E2 (Fig. 6a) and for each PEX (Fig. 6b and Supplementary Fig. 11). In addition to E2combo RNAi, the most striking effects on PEX regulation were found with RNAi for UBE2D1/2/3 and for UBE2G1, UBE2G2, and UBE2H, collectively considering both the number of regulated PEXs as well as the magnitude of changes induced (Fig. 6a). Conversely, other E2s (such as UBE2A/B, UBE2C, and UBE2E1/2/3) did not regulate any PEX (Fig. 6a). Cumulative analysis of PEX proteins indicated that many (but not all) PEXs are typically upregulated by RNAi for one or more E2s. For example, PEX2, PEX3, and PEX11A/B are upregulated by RNAi for several E2s whereas PEX5 and PEX7 are not (Fig. 6b). Altogether, these analyses indicate that certain E2s have prominent roles in PEX regulation and that likewise some peroxins are more pervasively modulated by E2 knockdown than others. In particular, UBE2D1/2/3 emerges as a prominent E2 family in PEX protein regulation. On this basis, we have further examined PEX modulation by these related E2s and found that siRNAs for UBE2D1/2/3 significantly increase the protein (but not the mRNA) levels for PEX1, PEX2, PEX3, PEX6, PEX11A, PEX11G, PEX12, PEX14, and PEX26 (Fig. 6c).

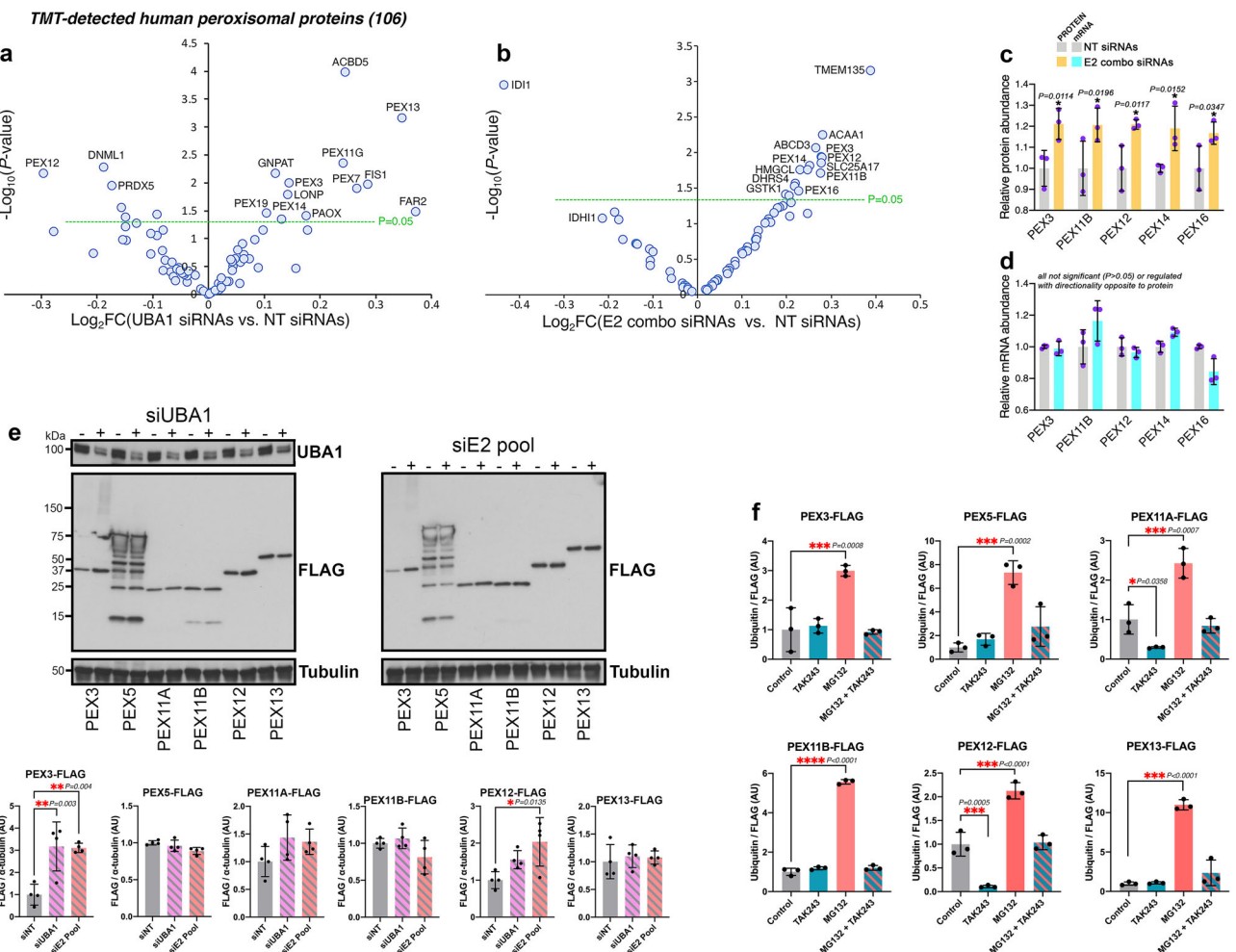

**Fig. 4 | Upregulation of peroxisomal proteins by UBA1/E2combo RNAi.** Modulation of the levels of peroxisomal proteins by UBA1 siRNAs (**a**) and E2combo siRNAs (**b**) compared to control NT siRNAs. The x-axis displays the Log₂FC of UBA1 RNAi (**a**) and E2combo RNAi (**b**) versus control NT RNAi whereas the y-axis reports the significance, indicated as -Log₁₀(P value; unpaired two-tailed t test). Overall, UBA1 and E2combo siRNAs upregulate the levels of several peroxisomal proteins, including peroxins (PEX). **c, d** PEX protein upregulation (P < 0.05 and Log₂FC > 0.2) upon E2combo knockdown occurs without significant and/or concordant mRNA changes. N = 3 biological replicates/group, mean ± SD (P values were obtained with unpaired two-tailed t tests for TMT data and with non-parametric ANOVA for RNA-seq data). **e** Expression of FLAG-tagged PEX proteins in HEK293T cells. UBA1 and E2combo RNAi increases the levels of PEX3-FLAG and PEX12-FLAG. N = 4 biological

replicates/group with mean ± SD and P values indicated (*P < 0.05 and **P < 0.01; one-way ANOVA). **f** PEX turnover occurs at least in part via the proteasome. Immunoprecipitation of FLAG-tagged versions of some of the PEX proteins that are upregulated by UBA1/E2combo RNAi (**a, b**). Anti-ubiquitin (Ub) immunoblotting is used to determine the ubiquitination status of FLAG-tagged PEX proteins. Compared to controls (gray), treatment with the proteasome inhibitor MG132 (orange) increases PEX ubiquitination and this is largely prevented by concomitant treatment with the UBA1 inhibitor TAK243 (orange-green stripes). Similar results are obtained for PEX3, PEX5, PEX11A, PE11B, PEX12, and PEX13; n = 3 biological replicates/group, mean ± SD, *P < 0.05 (one-way ANOVA). These findings indicate that peroxin turnover occurs at least in part via the proteasome. See also Supplementary Fig. 9. Source data are provided in the Source data file.

Besides peroxins, UBE2D1/2/3 siRNAs also increase the levels of other peroxisomal proteins, such as TMEM135 and ACBD3/5 (Fig. 6d).

We next sought to further test whether PEX protein upregulation induced by UBE2D1/2/3 siRNAs occurs independently from *PEX* gene transcription. To this purpose, HEK293T cells were transfected with FLAG-tagged PEX proteins not expressed under the control of their endogenous PEX promoters, and the PEX-FLAG protein levels were assessed by western blot (Fig. 6e). Knockdown of UBE2D1/2/3 increased the levels of PEX3-FLAG, PEX11A-FLAG, and to a lower extent of PEX12-FLAG, which correspond to PEX proteins upregulated by UBE2D1/2/3 RNAi in TMT studies (Fig. 6a–c). Conversely, the levels of PEX-FLAG proteins that were not modulated by UBE2D1/2/3 RNAi in TMT (PEX5-FLAG, PEX11B-FLAG, and PEX13-FLAG) were consistently not modulated when examined by western blot (Fig. 6e).

Having established an important role for UBE2D1/2/3 in modulating PEX protein levels, we next examined whether this PEX regulation entails UBE2D-dependent ubiquitination of PEX proteins. For

these studies, PEX- FLAG proteins were immunoprecipitated from cells treated with either UBE2D1/2/3 RNAi or control NT RNAi, and the ubiquitination status of PEX-FLAG determined with anti-ubiquitin antibodies. Compared to control NT siRNAs, knockdown of UBE2D1/2/3 reduced the ubiquitination of PEX13-FLAG and to a lower extent that of other PEX-FLAG proteins (Fig. 6f), suggesting that knockdown of UBE2D1/2/3 upregulates PEX protein levels by impeding their ubiquitination and turnover. Although PEX5 protein levels are not modulated by UBE2D1/2/3 siRNAs (Fig. 6a, b), we find that UBE2D1/2/3 siRNAs reduce PEX5 poly-ubiquitination (Fig. 6f). Previous studies have demonstrated that UBE2D/UbcH5 is responsible for PEX5 mono-ubiquitination and that this is key for PEX5 function as cargo receptor in peroxisomal protein import[47–50]. Our findings now suggest that UBE2Ds may also contribute to PEX5 poly-ubiquitination (Fig. 6f).

Because UBE2Ds modulate PEX protein levels, we next assessed whether UBE2D1/2/3 knockdown improves peroxisomal protein import as observed in response to cells with UBA1/E2combo

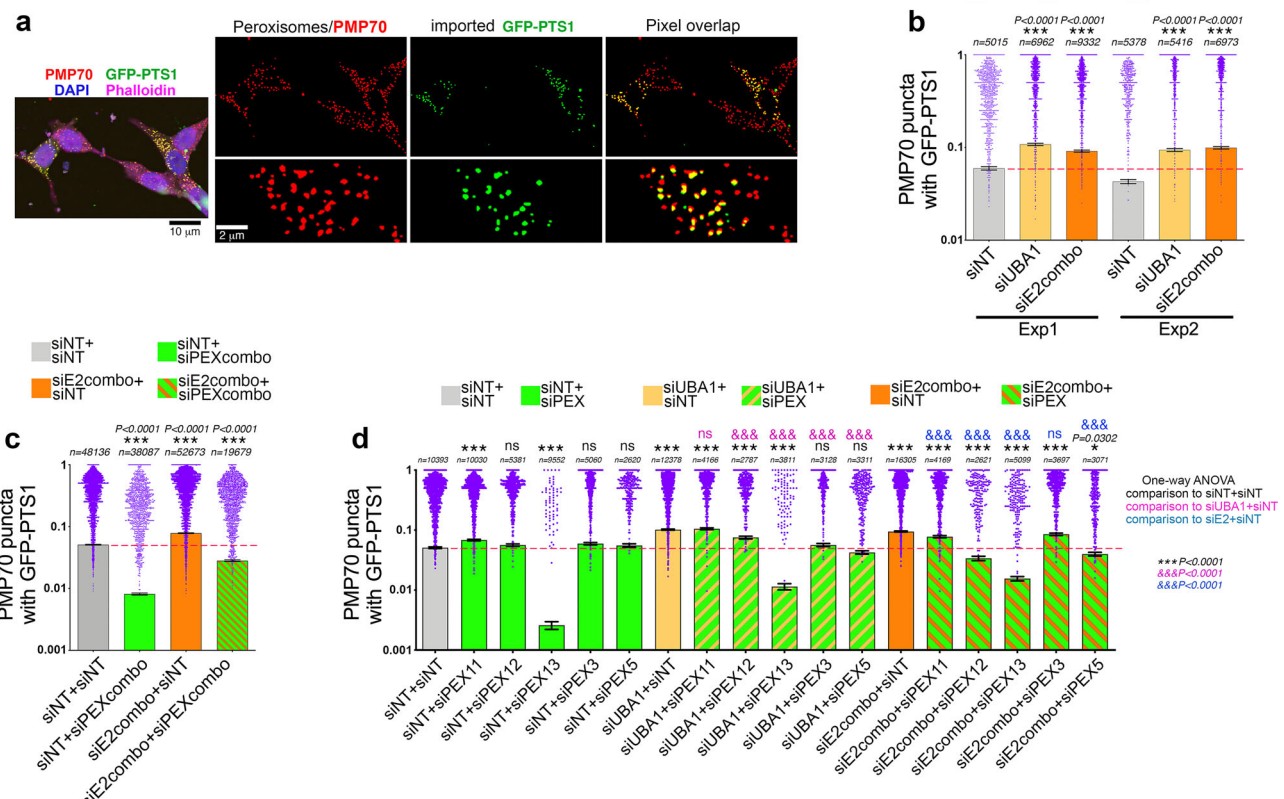

**Fig. 5 | UBA1/E2combo RNAi improves peroxisomal protein import in a PEX-dependent manner. a** HEK293T cells transfected with GFP-PTS1 (green) and immunostained for PMP70 (red) are segmented via machine-learning approaches to identify functional peroxisomes (GFP-PTS1+ and PMP70 + ; yellow) and ghost peroxisomes with defective PTS1-guided protein import (GFP-PTS1- and PMP70 + ; red). **b** RNAi for UBA1 (yellow) and for E2combo (orange) leads to a significant increase in the number of functional peroxisomes (PMP70 puncta with GFP-PTS1), compared to control cells treated with NT siRNAs (gray). **c**, **d** Knockdown of PEX proteins that are upregulated by UBA1/E2combo RNAi rescues back peroxisomal

protein import to levels closer to those found in control cells. Similar results are found with the concomitant RNAi of several PEX proteins (PEXcombo; **c**) and with RNAi for individual PEX proteins (**d**). In (**b**−**d**), n(peroxisomes)>5 × 10$^3$ (**b**), n > 1.9 × 10$^4$ (**c**), and n > 2.6 × 10$^3$ (**d**) with mean ±SEM indicated (the n for each condition are reported in the figure). The statistical significance was calculated by one-way ANOVA, and the significance values (***$P$ < 0.001, $^{\&\&\&}P$ < 0.001, and ns=not significant) are reported in black fonts (for comparisons to siNT+siNT), purple fonts (for comparisons to siUBA1+siNT), and blue fonts (for comparisons to siE2combo +siNT). Source data are provided in the Source data file.

knockdown (Fig. 5). To this purpose, HEK293T cells transfected with GFP-PTS1 were immunostained for PMP70 to identify functional peroxisomes (GFP-PTS1+ and PMP70 + ) and ghost peroxisomes with defective PTS1-guided protein import (GFP-PTS1- and PMP70+). These analyses indicated that UBE2D1/2/3 RNAi significantly increases the proportion of functional peroxisomes, as indicated by the increase in the PMP70 puncta with GFP-PTS1, and that this is prevented by PEX knockdown (Fig. 7a, b).

Altogether, these findings indicate that UBE2Ds are key E2s responsible for modulating PEX protein turnover and peroxisomal protein import.

### UBE2D modulates peroxisomal protein abundance and peroxisomal protein import in *Drosophila* skeletal muscle

We have found that the knockdown of UBA1 and a general decline in E2 function obtained with E2combo RNAi upregulates the levels of peroxins, and that this leads to increased peroxisomal protein import in human cells (Figs. 3−5). Interestingly, these effects are largely recapitulated by the knockdown of UBE2D1/2/3 (Figs. 6, 7), suggesting that this E2 family has a key role in this process. On this basis, we next examined whether UBE2Ds play a similar role in *Drosophila* skeletal muscle, a prominent tissue where the ubiquitin-proteasome system and E2 ubiquitin-conjugating enzymes have important roles[57,58,87].

While there are 4 highly related human UBE2Ds (i.e. UBE2D1, UBE2D2, UBE2D3, UBE2D4), effete (eff/CG7425) is the sole UBE2D

homolog in *Drosophila*[88]. On this basis, we examined whether eff knockdown regulates PEX protein levels, as observed for human cells with UBE2D1/2/3 knockdown. To this purpose, TMT mass spectrometry was utilized to identify protein changes induced in *Drosophila* skeletal muscle by knockdown of eff/UBE2D, by overexpression of human UBE2D2 (hUBE2D2), and by eff/UBE2D knockdown rescued by the concomitant expression of hUBE2D2, compared to controls (GFP RNAi and mcherry overexpression).

Analysis of the peroxisomal proteome indicates that overexpression of human UBE2D2 in *Drosophila* muscles significantly reduces Pex11 and Pex13 protein levels (Fig. 8a, d). Conversely, eff/UBE2D knockdown increases the abundance of several peroxisomal proteins including Pex11 and Pex13 (Fig. 8b, d) but not their mRNA levels (Supplementary Fig. 12), and this is prevented by concomitant hUBE2D2 overexpression compared to controls (Fig. 8c, d). However, other peroxisomal proteins that are upregulated by UBE2D/eff knockdown (such as Pex1) are not affected, presumably because human hUBE2D2 only partially compensates for the loss of *Drosophila* UBE2D/eff activity. Altogether, these findings indicate that UBE2D regulates peroxisomal protein abundance in *Drosophila* skeletal muscle (Fig. 8a−d) similar to what is observed in human cells with UBE2D1/2/3 knockdown (Fig. 6).

To test whether such an increase in Pex protein levels correspondingly increases peroxisomal protein import, *Drosophila* skeletal muscles that express eYFP with a PTS1 peroxisome targeting signal[89,90]

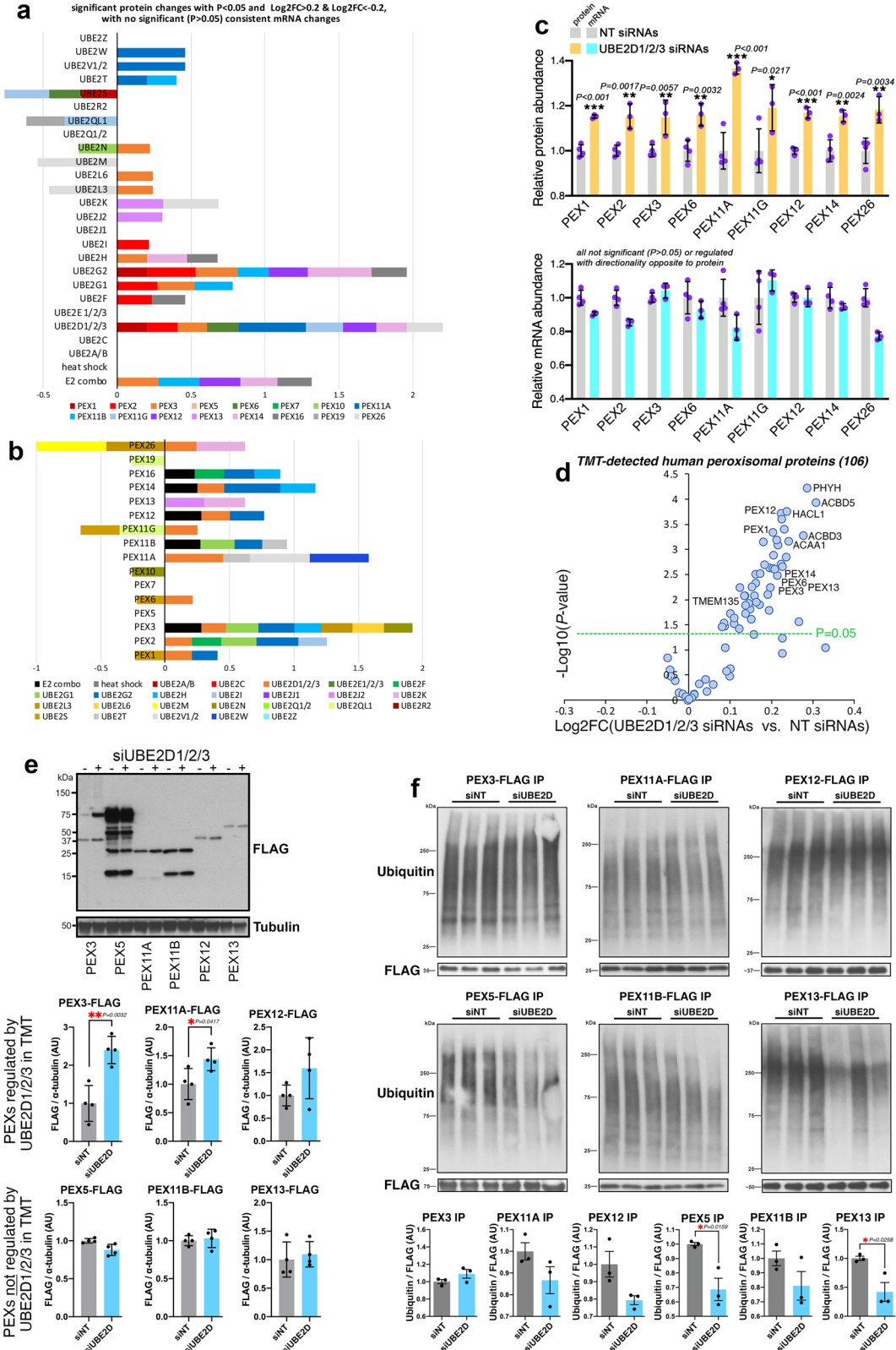

(eYFP-PTS1) were analyzed to identify peroxisomes with functional protein import (Fig. 8e, f). Compared to control RNAi, there was an increase in the number of peroxisomes with functional import in response to knockdown of UBA1 and of eff/UBE2D (Fig. 8g), whereas there were no changes in the size of peroxisomes (Fig. 8h). Altogether, these results indicate that knockdown of UBE2D/eff and UBA1 increases the number of peroxisomes with correctly imported eYFP-PTS1 in

*Drosophila* skeletal muscle (Fig. 8), indicating that this cellular adaptation occurs also in vivo as observed in cell culture (Figs. 4–7).

## Discussion

Ubiquitination is a fundamental mechanism for modulating protein function, localization, and abundance[4,91,92]. Despite the central role of ubiquitination in protein turnover, many proteins can be degraded in a

**Fig. 6 | UBE2D1/2/3 regulates the turnover of several peroxins (PEX) and UBE2D1/2/3 knockdown increases PEX protein levels post-transcriptionally. a** PEX protein upregulation by RNAi for individual or related E2s. Cumulative Log₂FC of PEX protein abundance induced by E2 RNAi versus control NT RNAi, arising independently from mRNA changes. UBE2D1/2/3 RNAi upregulates many PEX proteins, suggesting that UBE2D1/2/3 has a key role in PEX turnover. See also Supplementary Fig. 11. **b** PEX3 and PEX11 are among the PEXs more commonly upregulated upon E2 knockdown whereas others (such as PEX5 and PEX7) are not modulated. **c** UBE2D1/2/3 RNAi upregulates PEX proteins ($P < 0.05$ and Log₂FC > 0.2) without significant and/or consistent mRNA changes. N = 4 (NT siR-NAs) and N = 3 (UBE2D1/2/3 siRNAs) biological replicates/group, mean ±SD ($P$-values were obtained with unpaired two-tailed $t$ tests for TMT data and with non-parametric ANOVA for RNA-seq data). **d** Modulation of the peroxisomal proteome by UBE2D1/2/3 siRNAs. The $x$-axis displays the Log₂FC of UBE2D1/2/3 RNAi versus control NT RNAi whereas the $y$-axis reports the significance, -Log₁₀($P$ value).

UBE2D1/2/3 siRNAs upregulate several peroxins (PEX) and other peroxisomal proteins. **e** UBE2D1/2/3 knockdown increases PEX3-FLAG and PEX11A-FLAG protein levels, which correspond to PEXs that are upregulated by UBE2D1/2/3 RNAi in TMT studies (**a–c**). Conversely, PEX5-FLAG, PEX11B-FLAG, and PEX13-FLAG are not regulated by UBE2D1/2/3 RNAi, consistent with TMT studies (**a–c**). N = 4 biological replicates/group with mean ± SD and $P$ values indicated (*$P < 0.05$ and **$P < 0.01$, obtained with unpaired two-tailed $t$ tests). **f** Immunoprecipitation of PEX-FLAG proteins from cells treated with UBE2D1/2/3 RNAi (siUBE2D) and control NT RNAi (siNT). Anti-ubiquitin immunoblotting is used to determine the ubiquitination status of FLAG-tagged PEXs. Compared to control NT siRNAs (gray), knockdown of UBE2D1/2/3 (blue) reduces the ubiquitination of several FLAG-tagged peroxins, although this is significant only for PEX5-FLAG and PEX13-FLAG; n = 3 biological replicates/group, mean ± SEM, *$P < 0.05$ (unpaired two-tailed t-tests). Source data are provided in the Source data file.

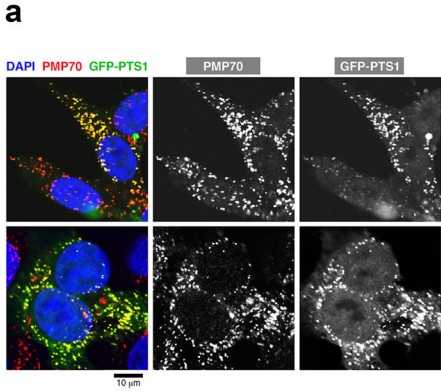

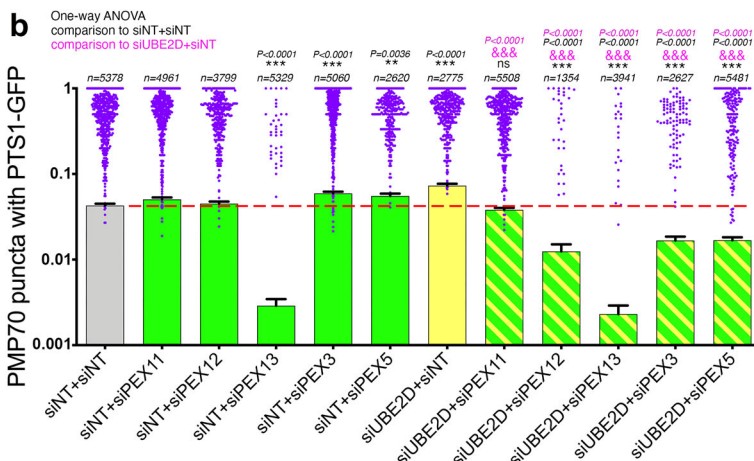

**Fig. 7 | Partial knockdown of UBE2D1/2/3 sustains peroxisomal protein import in human cells via compensatory PEX upregulation. a, b** Partial knockdown of UBE2D1/2/3 improves peroxisomal protein import in a PEX-dependent manner. **a** HEK293T cells transfected with GFP-PTS1 (green) and immunostained for PMP70 (red) to identify functional peroxisomes (GFP-PTS1+ and PMP70 + ; yellow) and ghost peroxisomes with defective PTS1-guided protein import (GFP-PTS1- and PMP70 + ; red). **b** UBE2D1/2/3 RNAi increases the proportion of functional

peroxisomes (PMP70 puncta with GFP-PTS1) and this is prevented by PEX knock-down. In (**b**), n(peroxisomes)>1.3 × 10³ with mean ±SEM indicated (the n for each condition are reported in the figure). The statistical significance was calculated by one-way ANOVA, and the significance values (**$P < 0.01$, ***$P < 0.001$, and &&&$P < 0.001$) are reported in black fonts (for comparisons to siNT+siNT) and purple fonts (for comparisons to siUBE2D+siNT). Source data are provided in the Source data file.

---

ubiquitination-independent manner by the proteasome[93], the autophagy/lysosome system[9,94–98] and by proteases/peptidases[99–102]. Conversely, because ubiquitination can be reversed via the action of de-ubiquitinating enzymes, E2/E3-mediated ubiquitination does not necessarily dictate the degradation fate of a target protein[93,103]. Therefore, an unresolved question is how ubiquitination, which is initiated by UBA1 and E2 enzymes, regulates protein abundance, and how this, in turn, affects cell function.

In this study, we have utilized deep-coverage TMT mass spectrometry to define how E2 ubiquitin-conjugating enzymes regulate protein abundance in human cells (Fig. 1). Our proteomic surveys identify protein subsets that are particularly sensitive to E2 perturbation, i.e. that are modulated by short-term (~3 days) knockdown of individual or related E2s. Specifically, by cross-comparing the proteins modulated by E2 siRNAs (as determined by deep-coverage TMT mass spectrometry) versus the gene expression changes induced by E2 siRNAs (defined via RNA-seq), we have determined the changes in protein levels that occur in response to E2 RNAi independently from transcriptional changes (Fig. 1 and Supplementary Figs. 2–4). While proteins upregulated by E2 knockdown likely result from decreased poly-ubiquitination and consequently reduced degradation, downregulated proteins may instead have decreased stability because of insufficient mono-ubiquitination, which regulates protein localization and function[4,91,92]. Interestingly, proteins modulated by E2 knockdown

span a wide range of lifetimes (Supplementary Fig. 4b), indicating that proteins most affected by E2 perturbation do not merely consist of short-lived proteins.

While we have examined cells with an experimental reduction in general and individual E2 function, there are several disease settings where UBA1/E2 function is reduced, such as VEXAS syndrome and spinal muscular atrophy X-linked 2 (SMAX2)[30,31,35,104]. Therefore, the proteomic changes identified in our experimental system may be relevant for disease pathogenesis. We have also defined a comprehensive network of E2 cross-interactions via IP-MS interactome mapping: these analyses have revealed extensive connections between different E2s and their physical association with other cellular complexes, including the proteasome (Fig. 2 and Supplementary Fig. 5, 6). Altogether, because E2s have a widespread expression (Supplementary Fig. 1), the integrated analyses reported in this study provide a framework to predict how global and individual perturbation of E2s rewires the proteome and impacts cellular function.

By determining the protein subsets that are modulated by a moderate decline in UBA1/E2 function, our proteomic surveys have also revealed changes in protein abundance that may underlie adaptive responses to reduced ubiquitination (Figs. 3–7). Several stress responses have been found to be triggered by diverse cellular insults, including perturbation of proteasome function and ubiquitin depletion[99,105–113]. A common theme that is shared by these and other

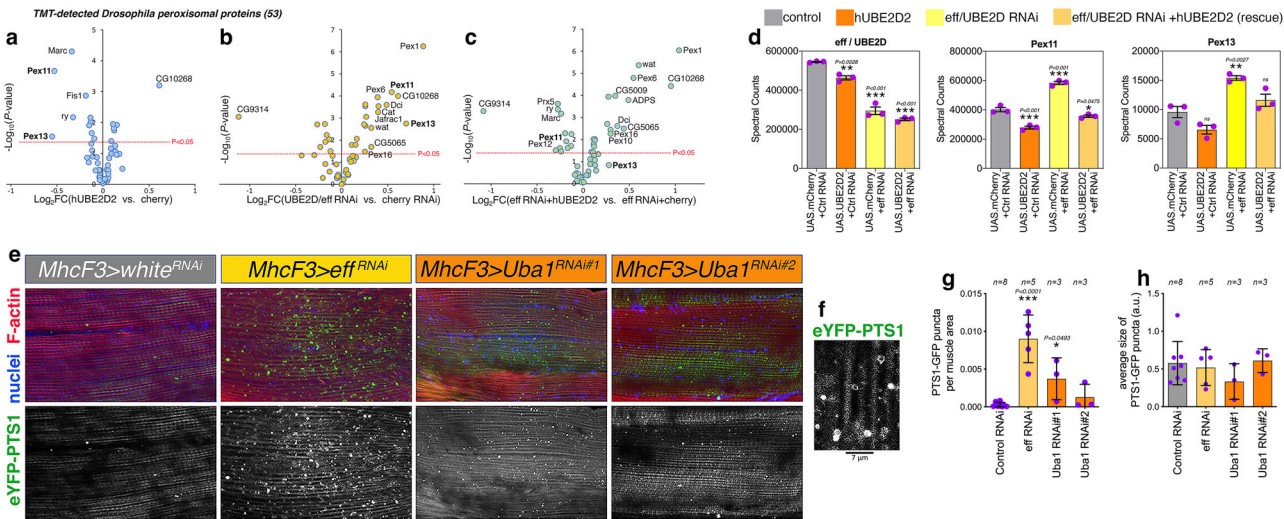

**Fig. 8 | UBE2D/eff knockdown upregulates PEX protein levels in *Drosophila* skeletal muscle.** TMT mass spectrometry identifies protein changes induced in *Drosophila* skeletal muscle by knockdown of eff (the sole *Drosophila* UBE2D1/2/3 homolog), by overexpression of human UBE2D2 (hUBE2D2), and by eff/UBE2D knockdown rescued by the concomitant expression of hUBE2D2, compared to controls (GFP RNAi and mcherry overexpression). **a** Overexpression of human UBE2D2 in *Drosophila* muscles has limited impact on the peroxisomal proteome, apart from a significant decline in Pex11 and Pex13 protein levels. **b** UBE2D/eff knockdown increases the abundance of peroxisomal proteins, including Pex11 and Pex13. **c** Rescue experiments with hUBE2D2 overexpression concomitant to UBE2D/eff knockdown, compared to mock rescue with cherry overexpression: hUBE2D2 overexpression prevents the upregulation of Pex11, Pex13, and other peroxisomal proteins by UBE2D/eff RNAi. However, some of the peroxisomal proteins that are upregulated by UBE2D/eff knockdown (e.g., Pex1) are not affected, presumably because human hUBE2D2 only partially compensates for *Drosophila* UBE2D/eff loss. In (a-c), the x-axis displays the Log₂FC whereas the y-axis reports the significance, -Log₁₀(P value). **d** Alongside a reduction in *Drosophila* eff/

UBE2D abundance, eff/UBE2D RNAi increases Pex11 and Pex13 protein levels (yellow) compared to controls (gray) but these are rescued by concomitant hUBE2D2 expression (light orange). Overexpression of hUBE2D2 by itself (dark orange) reduces Pex11 protein levels. N = 3 biological replicates/group with mean ±SEM, *$P < 0.05$, **$P < 0.01$, ***$P < 0.001$; ns= not significant (one-way ANOVA). In these comparisons, the number of UAS transgenes was kept equal to avoid Gal4 titration effects. **e, f** Immunostaining of *Drosophila* skeletal muscle with DAPI (to detect nuclei; blue), phalloidin (to detect F-actin; red), and eYFP with a PTS1 peroxisome targeting signal (eYFP-PTS1; green) to identify peroxisomes with functional protein import. **g** Knockdown of UBA1 and of eff/UBE2D increases the number of eYFP-PTS1+ puncta, indicative of functional peroxisomes, compared to control RNAi. **h** No changes in the size of peroxisomes are found. In (**g, h**), the n (biological replicates/group) is indicated in the figure, together with the mean ± SD, *$P < 0.05$ and ***$P < 0.001$ (one-way ANOVA). Altogether, these results indicate that the knockdown of UBE2D/eff and UBA1 increases the number of peroxisomes with correctly-imported eYFP-PTS1 in *Drosophila* skeletal muscle (e-h). Source data are provided in the Source data file.

cellular stress responses is that they rely on stress-sensing proteins that modulate the activity of effector signaling pathways, which in turn induce adaptive changes to re-establish homeostasis[114–124] (Fig. 9). By analyzing cells with decreased ubiquitination capacity (Figs. 3–7), we have found evidence for a novel type of adaptive response which is not based on stress-sensing and effector proteins but that rather depends on the ubiquitination status of E2 protein substrates (Fig. 9). This stress response does not re-establish the cellular capacity for ubiquitination but rather works towards preserving the function of organelles (such as peroxisomes) that require ubiquitination for their function and that are consequently challenged by a decline in UBA1/E2 activity. In particular, we find that the knockdown of UBA1, individual E2s, and of siRNAs for multiple E2s (E2 combo) upregulates the levels of PEX proteins (peroxins) necessary for peroxisomal protein import from the cytosol into the peroxisomal matrix (Figs. 3–7). Among the E2s that regulate PEX levels, we find that UBE2D1/2/3 knockdown elicited the upregulation of several PEX proteins (Fig. 6).

UBE2D (UbcH5) ubiquitin-conjugating enzymes have been previously found to mediate the mono-ubiquitination of the cargo receptor PEX5 and such modification is necessary for PEX5 cycling from the peroxisomal membrane to the cytosol and the initiation of a new round of protein import into the peroxisomal matrix[47–50]. Therefore, a predicted detrimental outcome of reduced ubiquitination is interference with the peroxisome-to-cytosol cycling of PEX5 and the consequent impediment of peroxisomal protein import. However, we find that this occurs concomitantly with decreased poly-ubiquitination and consequent upregulation of PEX 11, necessary for peroxisome biogenesis[47–49], and of other PEX proteins that are components of the

cargo receptor docking complex (importomer; Figs. 3–6), which sustains peroxisomal protein import by mediating the docking of the PEX5 cargo receptor to the peroxisomal membrane[47–50]. In addition to UBE2D RNAi, the knockdown of UBA1 and of several other E2s also increases PEX protein levels (Figs. 3–6), suggesting that reshaping peroxin abundance is a common response to maintain peroxisomal function in response to reduced ubiquitination. Likewise, PEX11/13 protein levels and the number of YFP-PTS1-positive, functional peroxisomes increase also in *Drosophila* skeletal muscle with UBA1 and UBE2D/eff knockdown (Fig. 8), indicating that PEX protein upregulation can also occur in vivo in response to a reduction in E2 function.

Although this study has primarily focused on the analysis of PEX proteins, we have found that other proteins that regulate organelle function are modulated by E2 knockdown, such as components of the mitochondrial electron transport chain (Fig. 3). The transcription-independent upregulation of electron transport chain proteins may thus sustain mitochondrial function in cells with reduced ubiquitination. Likewise, components of the endoplasmic reticulum, Golgi, and lysosome were also affected (Fig. 3), indicating that remodeling of other organelles apart peroxisomes likely takes place in cells with reduced E2 function.

Altogether, these findings indicate that perturbation of E2 function can be endured via built-in adaptive mechanisms that are based on decreased ubiquitination and consequent modulation of the levels of E2 protein substrates (Fig. 9). In particular, we have found that decreased poly-ubiquitination and consequent upregulation of E2 protein targets (i.e. PEX proteins of the cargo receptor docking complex) can in turn induce cellular processes (i.e. increased peroxisomal

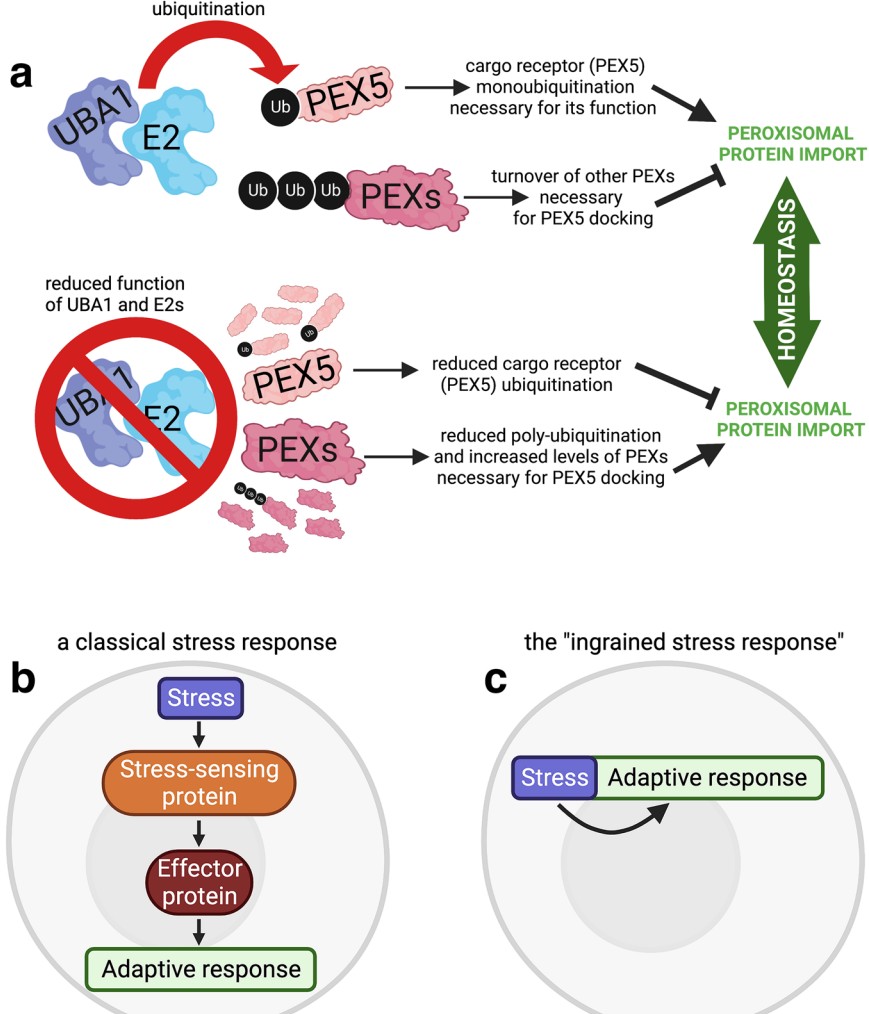

**Fig. 9 | An adaptive stress response induced by reduced ubiquitination and ingrained into the ubiquitin-dependent modulation of protein levels and function. a** Normally, the cargo receptor PEX5 binds in the cytosol to proteins with a PTS1 peroxisomal targeting signal type 1 and is responsible for their import into the peroxisomal matrix by binding to a cargo receptor docking complex (importomer) located on the peroxisomal membrane and composed by several other peroxins (PEXs). Mono-ubiquitination of PEX5 is known from previous studies to be mediated by the UBE2D ubiquitin-conjugating enzyme and to be key for the recycling of PEX5 back to the cytosol, where a new cycle of peroxisomal protein import can start. Consistently, UBE2D knockout has been reported to block PEX5 mono-ubiquitination and impede peroxisomal protein import. However, we find that a moderate decline in the levels of UBA1, of multiple E2s (E2 combo), and of UBE2D induces an adaptive response that paradoxically promotes peroxisomal protein import. Mechanistically, a moderate reduction in the cellular ubiquitination capacity and in UBE2D function compensates for reduced PEX5 ubiquitination by

decreasing the poly-ubiquitination and proteasomal degradation of peroxins that compose the importomer complex necessary for PEX5 docking to the peroxisomal membrane. Consequently, upregulation of PEXs of the importomer complex ensures peroxisomal protein import and maintains homeostasis. These findings indicate that perturbation of E2 function can be endured via built-in adaptive mechanisms that are based on decreased ubiquitination and consequent modulation of the levels of E2 protein substrates. **b** In a classic stress response, stress is detected by a sensor protein, which activates an effector protein or pathway, which in turn induces an adaptive response that protects from the initial stress. **c** A different type of adaptive response is induced by knockdown of the E1 ubiquitin-activating enzyme UBA1, by the general but partial reduction in the function of E2 ubiquitin-conjugating enzymes (E2combo RNAi), and by knockdown of the ubiquitin-conjugating enzyme UBE2D: this "ingrained stress response" depends on the decreased turnover and consequent upregulation of UBA1/UBE2D protein substrates, as explained in (**a**).

protein import) that compensate for decreased E2-mediated protein ubiquitination (i.e. PEX5 cargo receptor mono-ubiquitination, necessary for its cycling from the peroxisomal membrane to the cytosol; Fig. 9). Because of its inherent nature, ingrained in the E2 ubiquitin conjugation activity and impinging on E2 protein targets (Fig. 9), this type of adaptive response (here defined as the "ingrained stress response") appears to differ from other compensatory responses that rely on the activation of stress-sensing signaling pathways[114–117,119,121–130] (Fig. 9). In summary, this study provides a framework for understanding how cellular homeostasis is maintained in response to the moderate reduction in E2 function and the consequent rewiring of the proteome and organelle function.

## Methods

### Cell culture

HEK293T cells (ATCC #CRL-3216) were maintained at 37 °C with 5% $CO_2$ in high-glucose DMEM with glutamax (Gibco #10566016) containing 10% FBS (Gibco #10437-028) and penicillin/streptomycin (Gibco #15140122). For a typical experiment, around 50,000 cells/mL were plated in each well of a 12-well plate. The cells were transfected after they reached 80% confluence (typically the day after plating).

### *Drosophila* husbandry and fly stocks

Flies were maintained at 25 °C with 60% humidity on 12-h light/dark cycles with standard cornmeal/soy flour/yeast fly food[131,132]. The

following fly stocks were utilized: *Mhc-Gal4*[132,133], *MhcF3-Gal4*[58,134] (Bloomington stock center BL#38464), control *UAS-white^RNAi* (BL#33623), *UAS-eff^RNAi* (BL#35431), *UAS-Uba1^RNAi#1* (BL#25957), *UAS-Uba1^RNAi#2* (BL#76066), *UAS-eYFP-PTS1* (BL#64248), *UAS-mcherry* (BL#35787), *UAS-hUBE2D2* (BL#76819), and *UAS-mcherry^RNAi* (BL#35785). *Mhc-Gal4* was utilized to drive UAS transgene expression in Fig. 8a–d whereas *MhcF3-Gal4* was employed in Fig. 8e–g. After the appropriate crossing, the male progeny was analyzed at the age of ~2 weeks.

### Cell culture transfection

HEK293T cells (ATCC #CRL-3216) were maintained at 37 °C with 5% $CO_2$ in high-glucose DMEM with glutamax (Gibco #10566016) containing 10% FBS (Gibco #10437-028) and penicillin/streptomycin (Gibco #15140122).

For a typical experiment, around 50,000 HEK293T cells per mL were plated in each well of a 12-well plate. The cells were transfected after they reached 80% confluence (typically the day after plating). HEK293T cells were transfected with 50 µM siRNA targeting the specific gene or with control non-targeting (NT) siRNAs, using a ratio of 1 µL Lipofectamine2000 to 50 pmol of siRNA in 50 µL of OptiMEM medium. When combining multiple siRNAs together, the total amount of siRNA was kept constant to 50 pmol per µL of Lipofectamine, which was then added to 50 µL of OptiMEM and then to 1 mL of culture (for each well of a 12-well plate). The transfection medium was not removed but rather kept in culture for 3 days, equivalent to the time of treatment. The E2combo included all siRNAs targeting E2s, as indicated in Fig. 3c. The PEXcombo utilized in Fig. 5 included the siRNAs targeting PEX1, PEX11A, PEX11B, and PEX13.

In experiments where siRNAs were utilized alongside DNA plasmids, 1 µg of DNA was combined with 50 pmol of siRNAs per each 2 µL of Lipofectamine, which was then added to 50 µL of OptiMEM and then to 1 mL of culture (for each well of a 12-well plate). In all experiments, OptiMEM media was used for transfections and the cells were tested 3 days after transfection.

The following SMARTpool siRNAs (ON-TARGETplus siRNAs, Dharmacon/Horizon Discovery) were utilized to target human UBA1, E2s, and PEX proteins: UBA1 (L-004509-00-0005), UBE2A (L-009424-00-0005), UBE2B (L-009930-00-0005), UBE2C (L-004693-00-0005), UBE2D1 (L-009387-00-0005), UBE2D2 (L-010383-00-0005), UBE2D3 (L-008478-00-0005), UBE2E1 (L-008850-00-0005), UBE2E2 (L-031782-00-0005), UBE2F (L-009081-01-0005), UBE2G1 (L-010154-00-0005), UBE2G2 (L-009095-00-0005), UBE2H (L-009134-00-0005), UBE2I (L-004910-00-0005), UBE2J1 (L-007266-00-0005), UBE2J2 (L-008614-00-0005), UBE2K (L-009431-00-0005), UBE2L3 (L-010384-00-0005), UBE2L6 (L-008569-00-0005), UBE2M (L-004348-00-0005), UBE2N (L-003920-00-0005), UBE2Q1 (L-008631-00-0005), UBE2Q2 (L-008326-01-0005), UBE2Q2L (L-190154-00-0005), UBE2QL1 (L-024273-01-0005), UBE2R1 (L-003230-00-0005), UBE2R2 (L-009700-00-0005), UBE2S (L-009707-00-0005), UBE2T (L-004898-00-0005), UBE2V1 (L-010064-00-0005), UBE2V2 (L-008823-00-0005), UBE2W (L-009643-00-0005), UBE2Z (L-008596-00-0005), PEX1 (L-010331-00-0005), PEX3 (L-019544-00-0005), PEX5 (L-015788-00-0005), PEX11A (L-012622-00-0005), PEX11B (L-019520-00-0005), PEX12 (L-019337-00-0005), PEX13 (L-012591-00-0005), and control NT siRNAs (D-001810-10-05).

### Miscellaneous cell culture treatments and assays

MG132 (Sigma #M8699, ≥98% purity) and TAK-243 (Selleck #S8341, ≥99.6% purity), both dissolved in DMSO (Sigma #472301, ≥99.9% purity), were applied for 16 hours at a concentration of 50 µM and 20 nM, respectively, and compared to the corresponding DMSO dilution controls. Heat shock was done at 42 °C for 16 hours. Proteasome assays were done as previously described[58,135] by using the Proteasome-Glo 3-substrate assay system (Promega #G8531), which estimates chymotrypsin-like, trypsin-like, and caspase-like proteolytic activities.

### GFP-PTS1 and PMP70 immunofluorescence in HEK293T cells

HEK293T cells were seeded onto gelatin-coated (1% w/v) sterile glass plates (96-well glass-bottom plates, with #1.5 high-performance cover glass, 0.17±0.0005 mm: Cellvis #P96-1.5H-N; and 24-well glass-bottom plates: Cellvis #P24-1.5 P) and allowed to attach overnight. Cells were then co-transfected with siRNAs and with a plasmid (pEGFP-C1 + SKL) that expresses GFP-PTS1, a GFP targeted to peroxisomes (Addgene #53450). Three days after transfection, cells were fixed by adding 1 volume of fixative (PBS with 4% EM-grade PFA (Fisher Scientific #50-980-487), with no Triton X-100) to 1 volume of culture medium for 10 minutes at room temperature. Subsequently, the medium was removed, and the cells were incubated for an additional 10 minutes at room temperature with PBS + 0.1% Triton X-100. That medium was then removed, and the cells were blocked for 1 hour at room temperature with 1% BSA and 5% horse serum. The cells were then incubated overnight with primary antibodies (1:200) against GFP (Aves #GFP-1020) and PMP70 (abcam #ab3421) at 4 °C. Anti-FLAG antibodies (Cell Signaling #14793) were used in immunostaining for validating FLAG-tagged PEX proteins. After washes, the cells were incubated with DAPI (1:1000) together with secondary antibodies (1:200) for GFP (anti-chicken AlexaFluor488-conjugated) and PMP70 (anti-rabbit AlexaFluor555-conjugated) at room temperature for 2 hours. Imaging was done on a Nikon C2 confocal microscope. The cells were imaged directly in the Cellvis plates in PBS. Puncta identified by PMP70 fluorescence were defined as peroxisomes, and the overlap of PMP70 with PTS1-GFP was utilized to identify peroxisomes with optimal peroxisomal protein import. The co-immunostaining of HEK293T cells for PEX14 and PMP70 was done with the following primary antibodies: rabbit anti-PEX14 (ThermoFisher #10594-1-AP) and mouse anti-PMP70 (ab211533) at 1:200.

### Image analysis of HEK293T cells with peroxisomal markers

Nuclei (identified by DAPI staining) were segmented by using the StarDist deep learning segmentation package[136,137]. Channels with FITC and TRITC puncta (i.e. peroxisomes) were segmented by using a machine-learning approach based on ilastik[138]. Each punctum was labeled by using the scikit-image python package and the overlap between each punctum in the FITC and TRITC channels was calculated to define peroxisomes with functional versus dysfunctional import. The average number of peroxisomes per cell was estimated based on the number of PMP70-positive puncta normalized by the number of nuclei in a cell population.

### Protein sample preparation, protein digestion, and peptide isobaric labeling by tandem mass tags

For human cell culture samples, $1 \times 10^6$ HEK293T cells were seeded in a 10-cm petri dish and allowed to grow overnight before siRNA transfection. After 3 days, the cells (approximately $2 \times 10^6$) were washed 3 times with PBS, scraped, and the cell suspension split into two separate tubes (one for RNA-seq and one for TMT) and pelleted. Each cell pellet (approximately $1 \times 10^6$ cells) was then used for RNA or protein extraction. Each condition was analyzed with n = 3-4 biological replicates.

For the preparation of TMT samples, the cell pellet was extracted with 8 M urea lysis buffer (50 mM HEPES, pH 8.5, 8 M urea, and 0.5% sodium deoxycholate). Approximately 10 µg of protein was loaded into each gel lane. The protein concentration of the lysates was determined by Coomassie-stained short gels using bovine serum albumin (BSA) as standard[139]. The gel bands were excised and submitted for TMT. For solution-based samples, 100 µg of protein for each sample was digested with LysC (Wako) at an enzyme-to-substrate ratio of 1:100 (w/w) for 2 hours in the presence of 1 mM DTT. Following this, the samples were diluted to a final 2 M Urea concentration with 50 mM HEPES (pH 8.5), and further digested with trypsin (Promega) at an enzyme-to-substrate ratio of 1:50 (w/w) for at least 3 hours. The peptides were reduced by adding 1 mM DTT for 30 min at room

temperature (RT) followed by alkylation with 10 mM iodoacetamide (IAA) for 30 minutes in the dark at room temperature. The unreacted IAA was quenched with 30 mM DTT for 30 minutes. Finally, the digestion was terminated and acidified by adding trifluoroacetic acid (TFA) to 1%, desalted using C18 cartridges (Harvard Apparatus), and dried by speed vac. The purified peptides were resuspended in 50 mM HEPES (pH 8.5) and labeled with 16-plex Tandem Mass Tag (TMT) reagents (ThermoScientific) following the manufacturer's recommendations. For gel-based TMT samples, the short gel bands were washed twice with 50% acetonitrile and dried. The dried gel bands were incubated with trypsin at an enzyme-to-substrate ratio of 1:10 (w/w) for overnight digestion. Following the overnight digestion, the peptide solution from the short gel bands was extracted and dried down. The peptide mixture was resuspended in 50 mM HEPES (pH 8.5) and labeled with 16-plex Tandem Mass Tag (TMT) reagents (Thermo-Scientific) following the manufacturer's recommendations.

For TMT of *Drosophila* samples, 50 thoraces (consisting primarily of skeletal muscle) from 2-weeks-old male flies were collected and homogenized in 8 M urea lysis buffer (50 mM HEPES, pH 8.5, 8 M urea). After homogenization with zirconium beads in a NextAdvance bullet blender, 0.5% sodium deoxycholate was added to the tissue homogenates, which were then pelleted to remove fly cuticle debris. The resulting supernatant was further prepared for TMT mass spectrometry as described above.

## Two-dimensional HPLC and mass spectrometry

The TMT-labeled samples were mixed equally, desalted, and fractionated on an offline HPLC (Agilent 1220) using basic pH reverse-phase liquid chromatography (pH 8.0, XBridge C18 column, 4.6 mm × 25 cm, 3.5 µm particle size, Waters). The fractions were dried and resuspended in 5% formic acid and analyzed by acidic pH reverse phase LC-MS/MS analysis. The peptide samples were loaded on a nanoscale capillary reverse phase C18 column (New objective, 75 um ID × ~25 cm, 1.9 µm C18 resin from Dr. Maisch GmbH) by an HPLC system (Thermo Ultimate 3000) and eluted by a 60-min gradient. The eluted peptides were ionized by electrospray ionization and detected by an inline Orbitrap Fusion mass spectrometer (Thermo-Scientific). The mass spectrometer is operated in data-dependent mode with a survey scan in Orbitrap (60,000 resolution, $1 \times 10^6$ AGC target and 50 ms maximal ion time) and MS/MS high-resolution scans (60,000 resolution, $2 \times 10^5$ AGC target, 120 ms maximal ion time, 32 HCD normalized collision energy, 1 $m/z$ isolation window, and 15 s dynamic exclusion).

## MS data analysis

The MS/MS raw files were processed by the tag-based hybrid search engine JUMP[140]. The raw data were searched against the UniProt human and *Drosophila* databases concatenated with a reversed decoy database for evaluating false discovery rates. Searches were performed using a 15-ppm mass tolerance for both precursor and product ions, fully tryptic restriction with two maximal missed cleavages, three maximal modification sites, and the assignment of $a$, $b$, and $y$ ions. TMT tags on Lys and N-termini (+304.20715 Da) were used for static modifications and Met oxidation (+15.99492 Da) was considered as a dynamic modification. Matched MS/MS spectra were filtered by mass accuracy and matching scores to reduce the protein false discovery rate to ~1%. Proteins were quantified by summing reporter ion intensities across all matched PSMs using the JUMP software suite[141]. Analysis of linkage-specific ubiquitination was done with JUMPptm[51] and is reported in Supplementary Data 2. Categories enriched in protein sets were identified with DAVID[142].

The TMT mass spectrometry proteomics data are reported in Supplementary Data 1 and have been deposited to the ProteomeXchange Consortium via the PRIDE partner repository with the following dataset identifiers:

PXD042303 (Proteomic changes induced by knockdown of E2 ubiquitin-conjugating enzymes (UBE2A + B; UBE2C; UBE2D1 + D2 + D3; UBE2E1 + E2) in human HEK293 cells versus controls).

PXD042331 (Proteomic changes induced by knockdown of E2 ubiquitin-conjugating enzymes (UBE2F; UBE2G1; UBE2G2; UBE2H) in human HEK293 cells versus controls).

PXD042333 (Proteomic changes induced by knockdown of E2 ubiquitin-conjugating enzymes (UBE2I; UBE2J1; UBE2J2; UBE2K) in human HEK293 cells versus controls).

PXD042337 (Proteomic changes induced by knockdown of E2 ubiquitin-conjugating enzymes (UBEL3; UBE2L6; UBE2N; UBE2M) in human HEK293 cells versus controls).

PXD042339 (Proteomic changes induced by knockdown of E2 ubiquitin-conjugating enzymes (UBE2Q1 + Q2; UBE2QL1; UBE2R2; UBE2S) in human HEK293 cells versus controls).

PXD042340 (Proteomic changes induced by knockdown of E2 ubiquitin-conjugating enzymes (UBE2T; UBE2V1 + V2; UBEW; UBE2Z) in human HEK293 cells versus controls).

PXD042341 (Proteomic changes induced by the simultaneous knockdown of multiple E2 ubiquitin-conjugating enzymes (E2pool) in control and heat-shocked human HEK293 cells versus controls).

PXD042345 (Proteomic changes induced by knockdown of UBE2D/eff in adult *Drosophila* skeletal muscle, rescue with human UBE2D2, and controls).

PXD042347 (Proteomic changes induced by knockdown of the E1 ubiquitin-activating enzyme UBA1 in human HEK293 cells versus controls).

## RNA sequencing

Samples for RNA sequencing were prepared in parallel with samples for protein extraction, as explained in the corresponding method section. Cell pellets were lysed with TRIzol (Ambion #15596018) and the RNA extracted by isopropanol precipitation from the aqueous phase. RNA sequencing libraries for each sample were prepared from 1 µg total RNA by using the Illumina TruSeq RNA Sample Prep v2 Kit per the manufacturer's instructions, and sequencing was completed on the Illumina NovaSeq 6000. The 100-bp paired-end reads were trimmed, filtered against quality (Phred-like Q20 or greater) and length (50-bp or longer), and aligned to the human reference genome GRCh38/hg38 by using CLC Genomics Workbench v12.0.1 (Qiagen). For gene expression comparisons, we obtained the TPM (transcript per million) counts from the CLC RNA-Seq Analysis tool. The differential gene expression analysis was performed by applying the non-parametric ANOVA using the Kruskal-Wallis and Dunn's tests on log-transformed TPM between 3 to 4 replicates of experimental groups, implemented in Partek Genomics Suite v7.0 software (Partek Inc.). The RNA-seq data discussed in this publication have been deposited in the NCBI's Gene Expression Omnibus (GEO) with accession number GSE222110.

(RNA-seq data of HEK293 cells treated with siRNAs for individual or related human E2 ubiquitin-conjugating enzymes (UBE2A + B; UBE2C; UBE2D1 + D2 + D3; UBE2E1 + E2; UBE2F; UBE2G1; UBE2G2; UBE2H; UBE2I; UBE2J1; UBE2J2; UBE2K; UBEL3; UBE2L6; UBE2N; UBE2M; UBE2Q1 + Q2; UBE2QL1; UBE2R2; UBE2S; UBE2T; UBE2V1 + V2; UBEW; UBE2Z), a pool of siRNAs targeting all E2s (E2combo), siRNAs targeting the E1 ubiquitin-activating enzyme UBA1, and controls).

## qRT-PCR

For qRT-PCR, cDNAs were reverse transcribed with the iScript cDNA synthesis kit (Bio-Rad #1708840) from 500 ng total RNA. qRT-PCR was performed by using the IQ Sybr Green supermix (Bio-Rad #170-8885). *GAPDH* and *Tub84B* were utilized as normalization respectively for human and *Drosophila* samples[132,143,144]. Supplementary Data 5 reports the qRT-PCR oligos used for human and *Drosophila* samples.

### Cloning of FLAG-tagged E1s and E2s

For cloning E1s (UBA1, UBA2, UBA3, UBA5, UBA6, ATG7, NAE1, SAE1) and E2s (UBE2A, UBE2B, UBE2C, UBE2D1, UBE2D2, UBE2D3, UBE2E1, UBE2E2, UBE2F, UBE2G1, UBE2G2, UBE2H, UBE2I, UBE2J1, UBE2J2, UBE2K, UBE2L3, UBE2L6, UBE2M, UBE2N, UBE2Q2, UBE2QL1, UBE2R1, UBE2R2, UBE2S, UBE2T, UBE2U, UBE2V1, ATG3, and ATG10) with a FLAG tag, these were amplified by PCR from HEK293T cDNA and cloned into the pCDH-EF1-T2A-GFP vector (SBI #CD527A-1). E1/2s with a C-terminal FLAG tag were cloned downstream of the EF1 promoter, and the C-terminal FLAG tag was followed by a stop codon to ensure that no GFP is produced. This cloning resulted in 38 pCDH-EF1-E1/E2-FLAG-STOP (T2A-GFP) plasmids. Supplementary Data 6 reports the oligos utilized for cloning.

### Co-immunoprecipitation experiments

HEK293T cells were cultured and transfected with plasmids encoding for FLAG-tagged E1/E2s and PEX proteins by following the methods described above. Plasmids for expression of FLAG-tagged PEX proteins were obtained from LSBio: human PEX11A (LS-N39529), human PEX11B (LS-N51398), human PEX5 (LS-N51726), human PEX3 (LS-N40426), human PEX13 (LS-N55089), and mouse PEX12 (LS-N141352) in the pCMV3 expression vector. Plasmids for expression of FLAG-tagged E1 and E2 proteins were obtained by cloning as described above.

Cells were lysed with the NP40 cell lysis buffer (Invitrogen #FNN0021) with protease inhibitors, sonicated for 5 s with a Branson Digital Sonifier at 30% amplitude, centrifuged, and the protein content of the supernatant measured by using the Bio-Rad protein assay. 100 µg of total protein for all samples was added to a final volume of 200 µL of NP40 cell lysis buffer. A small aliquot (20 µL) was removed and prepared for blotting as input. The remaining protein sample was mixed with 50 µL of FLAG affinity resin (Fisher Scientific #15895833) and incubated in rotating microcentrifuge tubes at 4 °C overnight. The mix was centrifuged at 2000 g for 2 minutes and the supernatant was carefully removed, leaving the affinity resin and co-immunoprecipitated proteins that were then washed 3 times with PBS and once with NP40 cell lysis buffer. The bait and co-immunoprecipitating proteins were then eluted with 20 µL of 5 µg/mL FLAG peptide (Sigma #F3290) in NP40 cell lysis buffer while rocking at room temperature. SDS blue loading buffer and DTT were added to the input and eluted protein supernatants, and heated at 95 °C for 5 min for SDS-PAGE.

### Spectral counting for co-immunoprecipitation experiments

Following the co-immunoprecipitation procedures above, the samples were run for a short distance on SDS-polyacrylamide gels. The proteins in the gel bands were reduced with DTT to break disulfide bonds and the Cys residues were alkylated by iodoacetamide. The gel bands were then washed, dried down in a speed vacuum, and rehydrated with a buffer containing trypsin for overnight proteolysis. The digested samples were acidified, and the peptides were extracted multiple times. The extracts were pooled, dried down, and reconstituted in a small volume. The peptide samples were loaded onto a nanoscale capillary reverse phase C18 column by a HPLC system (Thermo EasynLC 1000), and eluted by a gradient for ~90 min. The eluted peptides were ionized by electrospray ionization and detected by an inline mass spectrometer (Thermo Elite). The MS spectra were collected, and the 20 most abundant ions were sequentially isolated for MS/MS analysis. This process was cycled over the entire liquid chromatography gradient.

Database searches were performed with the Sequest search engine included in the in-house JUMP software package. All matched MS/MS spectra were filtered by mass accuracy and matching scores to reduce the protein false discovery rate to ~1%. Finally, all proteins identified in one gel lane were combined. The total number of spectra, i.e. spectral counts (SC), matching individual proteins reflect their relative abundance in one sample after normalization for protein size. The spectral counts were used for the calculation of $P$ values to identify significantly enriched proteins[57,145]. A Significance Analysis of INTeractome (SAINT) score was calculated to indicate the probability that the detected proteins are more abundant for that FLAG-tagged bait compared to a negative control in which a lysate from mock-transfected cells that do not express any FLAG-tagged protein underwent the same co-immunoprecipitation procedures[146].

A SAINT ≥ 0.65 was classified as a significantly enriched interactor compared to control samples. Cytoscape was utilized to represent protein-protein interaction data, which consisted of 1171 significantly interacting pairs for 30 E2 baits. Based on the whole proteome mass spectrometry data from HEK293T cells, the average spectral counts for proteins detected by affinity purification were substantially higher than the average for all detected proteins (Supplementary Fig. 5). Therefore, the identification of interacting proteins is limited by the abundance and relative ease of MS-based detection of these proteins. In addition to SAINT, we also compared detected proteins with the Contaminant Repository for Affinity Purification (CRAPome)[60]. Additionally, we compared maximum spectral counts for proteins detected in association with E2 baits and found that these were consistently higher than those from control (no bait) purifications, indicating that many detected proteins are bona fide interactors and are unlikely to be contaminants of the affinity purification process (Supplementary Fig. 5). The IP-mass spectrometry interactome data are reported in Supplementary Data 3-4 and have been deposited to the ProteomeXchange Consortium via the PRIDE partner repository with the dataset identifier PXD042361, (Interactome of E1 ubiquitin-activating enzymes and of E2 ubiquitin-conjugating enzymes in human HEK293 cells).

### Standard western blot analyses

Western blots were done as previously described according to routine protocols[147,148]. For the analysis of FLAG-tagged PEX protein levels, cells were harvested in NP40 lysis buffer (Invitrogen #FNN0021), analyzed by SDS-PAGE (Bio-Rad #4561096), followed by probing with anti-FLAG (Cell Signaling #14793) and anti-α-tubulin (Cell Signaling #2125) antibodies at 1:1000.

### Western blots of detergent-soluble and insoluble protein fractions from HEK293T cells

Detergent-soluble and insoluble fractions from HEK293T cells were obtained following the procedures described before[135,149] with slight modifications. Specifically, HEK293T cells were lysed in NP40 lysis buffer (Invitrogen #FNN0021) with protease inhibitors by gentle pipetting. The cell homogenate was centrifuged at 21000 x $g$ and the supernatant was retrieved (detergent-soluble fraction). The remaining pellet was washed twice in ice-cold NP40 lysis buffer and then resuspended in urea buffer (RIPA buffer with 8 M urea). The samples were then centrifuged at 21000 x $g$ for 10 min to pellet any remaining cell debris. The resulting supernatant (detergent-insoluble fraction) was retrieved, boiled with SDS-Blue loading buffer (Cell Signaling #7722) and DTT (Cell Signaling #1425 S), and analyzed by SDS-PAGE and western blot. The following antibodies were utilized at 1:1000 to examine detergent-soluble and/or detergent-insoluble fractions: anti-UBA1 (Cell Signaling #4891), anti-ubiquitin (P4D1, Santa Cruz Biotechnology #sc-8017), anti-K27 linkage-specific ubiquitination (abcam #ab238442), anti-K48 linkage-specific ubiquitination (Cell Signaling #8081), anti-K63 linkage-specific ubiquitination (Cell Signaling #5621), anti-NEDD8 (Cell Signaling #2754), anti-SUMO1 (Cell Signaling #4930), anti-SUMO2/3 (Cell Signaling #4971), and anti-α-tubulin (Cell Signaling #2125).

## Miscellaneous data analyses

For the cross-comparison of protein changes induced by RNAi of E2s, the starting point was a table with the proteins that were detected across all TMT experiments that have profiled the proteomic changes induced by the knockdown of individual/related E2s. Proteins with significant modulation ($P < 0.05$ and a $Log_2R$ change of $>0.2$ and $<-0.2$) by at least one E2 knockdown compared to control NT siRNAs were selected for further analysis. Subsequently, the $Log_2R$ proteomic changes induced by the knockdown of a given E2 (normalized to the control NT siRNAs) were cross compared with those induced by the knockdown of another E2: $R^2$ values corresponding to these pairwise comparisons were obtained in Excel. Subsequently, a similarity score was computed for each pairwise comparison by multiplying each $R^2$ value by the number of regulated proteins detected in that pairwise comparison.

To avoid batch-derived biases in estimating the degree of similarity of proteomic changes, intra-batch z-scores were calculated by cross-comparing E2-induced proteomic changes obtained within the same TMT set. For calculating intra-batch z-scores, we utilized the mean and standard deviation values calculated from the similarity scores obtained from cross-comparisons within the same TMT set. For all other comparisons between proteomic changes induced by E2 knockdown datasets obtained from distinct TMT batches, z-scores were obtained by utilizing the mean and standard deviation calculated from the similarity values obtained from all extra-batch cross-comparisons (i.e. similarity values outside of dashed boxes in Supplementary Fig. 6a). Z-scores were calculated as (individual similarity value – mean of similarity values in the group) / group standard deviation.

For the cross-comparison of E2 physical interactions obtained via IP/MS, SAINT scores relative to the interactome of each E2 were obtained from the spectral counts after filtering out aspecific interactions, as defined based on the CRAPome database[60]. The interactome of each E2 was cross compared with that of another E2: an $R^2$ value was obtained for each pairwise comparison of SAINT scores.

The human peroxisomal proteome was retrieved primarily from the peroxisome database (http://www.peroxisomedb.org/), with a few additions from Gronemeyer et al.[72]. The *Drosophila* peroxisomal proteome was inferred based on sequence homology via DIOPT searches (https://www.flyrnai.org/cgi-bin/DRSC_orthologs.pl). Schemes were drawn with BioRender.

## Immunostaining and laser scanning confocal microscopy of *Drosophila* skeletal muscle

*Drosophila* indirect flight skeletal muscles were immunostained according to the procedures previously reported[58,148,150]. Specifically, whole flies were frozen on glass slides with OCT in liquid nitrogen and then bisected longitudinally at the median plane to obtain hemithorax sections, which were then fixed with 4% paraformaldehyde and 0.1% Triton X-100 for 30 min, washed, and then incubated with 1:200 dilutions of primary and secondary antibodies. Chicken anti-GFP (Aves #GFP-1010) primary antibodies were used followed by AlexaFluor488-conjugated anti-chicken secondary antibodies (ThermoFisher #A34054) together with DAPI (ThermoFisher #D1306, 1:1000) and AlexaFluor635-conjugated phalloidin (Life Technologies #A22284, 1:100).

Stained thoraces were mounted on glass slides by polyvinyl-alcohol with DABCO (Sigma) and #1.5 coverslips and imaged by using a Nikon C2 confocal microscope. NIH ImageJ was used to quantify the number and average size of peroxisomes marked by eYFP-PTS1.

## Quantification and statistical analysis

Data organization, scientific graphing, and statistical analyses were done with Microsoft Excel (version 14.7.3) and GraphPad Prism (version 8). The unpaired two-tailed Student's t-test was used to compare the means of two independent groups to each other. One-way ANOVA with post-hoc testing was used for multiple comparisons of more than two groups of normally distributed data. Two-way ANOVA with post-hoc testing was used for multiple comparisons of more than two groups of normally distributed data in the presence of two independent variables. The n for each experiment can be found in the figures and represents independently generated samples. Bar graphs represent the mean ± SD or the mean ± SEM, as specified in the figure legend. A significant result was defined as $P < 0.05$. Throughout the figures, asterisks and ampersand symbols indicate the significance of $P$ values: $*P < 0.05$, $**P < 0.01$, and $***P < 0.001$. Statistical analysis of RNA-seq and mass spectrometry data is described in the corresponding method sections.

## Data availability

The data supporting the findings of this study are available within the paper, Supplementary Figures 1–12, Supplementary Data 1-6, and the Source Data file. The TMT mass spectrometry proteomics data have been deposited to the ProteomeXchange Consortium via the PRIDE partner repository and are accessible with the dataset identifiers PXD042303, PXD042331, PXD042333, PXD042337, PXD042339, PXD042340, PXD042341, PXD042345, and PXD042347. The IP-mass spectrometry interactome data have been deposited to the ProteomeXchange Consortium via the PRIDE partner repository with the dataset identifier PXD042361. The RNA-seq data have been deposited at the Gene Expression Omnibus (GEO) with the identifier GSE222110. There are no restrictions in the availability of the data and tools generated by this study. Source data are provided with this paper.

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

## Acknowledgements

We thank the Light Microscopy facility and the Hartwell Center for Bioinformatics and Biotechnology at St. Jude Children's Research

Hospital. *Drosophila* stocks were provided by the Bloomington stock center. We also thank Dr. Andrew Simmonds for the kind gift of antibodies, and Addgene for plasmids. Work in the Demontis lab is supported by the National Institute on Aging of the NIH (R01AG055532 and R21AG079267) and by the Alzheimer's Association (AARG-NTF-22-973220). The Peng lab is partially supported by the NIH (RF1AG068581). The content is solely the responsibility of the authors and does not necessarily represent the official views of the National Institutes of Health. Research at St. Jude Children's Research Hospital is supported by the ALSAC.

## Author contributions

L.H. did most of the experiments and data analysis, with help from A.St. and F.A.G.; V.P. did all TMT mass spectrometry and corresponding data analyses, with help from K.Ka.; L.H. and V.P. contributed equally as first authors; Y.F., S.P., Y.L., and X.W. performed computational analyses of mass spectrometry data with JUMPptm; K.Ko. did IP/MS analyses; B.X., X.W., and H.T. performed additional analyses; M.C. assisted with Drosophila experiments; A.Sh. analyzed confocal microscopy images; Y-D.W. analyzed RNA-seq data; J.P. supervised the mass spectrometry studies; F.D. supervised the project and wrote the manuscript.

## Competing interests

The authors declare no competing interests.
