## [Peer Review File · Nature Communications]

REVIEWER COMMENTS

Reviewer #1 (Remarks to the Author):

Comments to the Author:

In this manuscript, Hunt and colleagues utilized quantitative proteomics and functional genomics to characterize the perturbation of cellular proteome upon decreased ubiquitination. The authors found that the knockdown of UBA1 ubiquitin-activating enzyme and E2 ubiquitin-conjugating enzymes reshapes the proteome, and integrated analyses reveal specific E2 target proteins. Intriguingly, the knockdown of either UBA1 or E2 enzymes increases peroxins and peroxisomal import function. Thus, the work uncovers a novel interplay between the protein ubiquitination system and peroxisomal import function. I have a few concerns as listed below.

1. In Figure 4, the authors show that UBA1/E2combo RNAi increases the levels of PEX3, PEX11A and PEX12. Except for PEX12, PEX3, and PEX11A are not directly involved in the regulation of peroxisomal import. PEX3 mainly works with PEX19 in controlling peroxisome de novo biogenesis, while PEX11 regulates peroxisome fission. The increased levels of PEX3 and PEX11A might result in a higher number of peroxisomes. The authors should check this possibility by quantifying the number of PMP70-positive puncta in UBA1/E2combo knockdown cells. PEX14 antibody should also be used to validate PMP70 immunostaining results.

In Figure 5d, the knockdown of either PEX3 or PEX11 did not significantly block UBA1/E2combo KD-induced peroxisomal import. This is likely because the two proteins are not directly involved in peroxisome import. But the authors may examine the requirement of PEX3 or PEX11 for UBA1/E2-regulated peroxisome de novo biogenesis.

2. In Figure 5f-5g, the authors performed lipidomics and identified ~100 lipid species that are modulated by UBA1 and E2 in a peroxin-dependent manner. However, it is unclear whether the changes in lipid profiles are due to enhanced peroxisome function or due to altered metabolism in other organelles like ER and mitochondria. Because it is known that peroxisomes interact with many organelles to maintain cellular homeostasis, it would be nice to show what lipid species are directly linked to peroxisome-specific metabolism (like VLCFA beta-oxidation), and which lipids are related to ER or mitochondrial metabolism.

3. In Figure 6, I wonder if the regulation of PEX ubiquitination by UBE2D1/2/3 is direct. Can authors perform a Co-IP analysis to examine the possible interaction between UBE2D1/2/3 and individual peroxin?

In Figure 6d, I guess that "UBE2D knockdown" was missed from the figure panel labels.

4. In Figure 6f (and the model Figure 7k), the authors indicate that PEX5 ubiquitination is downregulated by UBE2D KD. However, it is unclear whether both mono-ubiquitination and poly-ubiquitination of PEX5 are reduced. If mono-ubiquitination of PEX5 is reduced, one would expect to find impaired peroxisomal import. The authors need carefully examine mono-ubiquitination vs. poly-ubiquitination of PEX5 upon UBE2D KD.

5. In Figure 7, I suggest the authors take a look at the peroxisome number in fly muscle as well, for the same reason mentioned above.

Reviewer #2 (Remarks to the Author):

Protein ubiquitination is one of the major post-translational modifications in cells regulating most, if not all, cellular processes. Ubiquitination has various outcomes including modulation of protein stability, localization, function, and degradation. How cells respond to an overall inhibition of ubiquitinylation remains elusive. In this manuscript the authors explore the impact of UBA1/E2s knockdown on the cellular proteome using quantitative proteomics. They found that specific compartments are more affected by a moderate, partial reduction in ubiquitin conjugation, such as peroxisome, ER and mitochondria. They further define a network of E2 cross-interaction via IP-MS interactome mapping. Having found a more profound effect of UBA1/E2s knockdown on peroxisomal proteins, they embarked on defining the underlying mechanisms. They identified that UBA1/E2 knockdown boosts peroxisomal import via the counterbalancing upregulation of peroxins necessary for PEX5 docking to the peroxisomal membrane using both HEK293T cells and drosophila. They argue that homeostatic mechanisms are triggered to preserve organelle homeostasis upon reduction of overall ubiquitinylation.

Better understanding how cells adapt to ubiquitin conjugation defect along with its impact on proteostasis is an important and timely contribution. Data presented in the manuscript support overall authors' conclusion, although some specific conclusions are less convincing (see below). On the whole, the paper is well written and interesting but may have benefited from being more focused, as some data don't bring much to the story (e.g., IP-MS interactome mapping found only few E3s co-immunoprecipitated with E2s and those E3s were not further studied in the paper). In summary, I do think that this paper will be a good addition to the ubiquitin and proteostasis field and I would support publication of this manuscript providing the more specific concerns listed below are addressed.

Major comments:

1. Fig. 2f shows the overall E2 interactome, do the authors know why only ~40 E3/DUBs are identified out of the ~620-800 E3/DUB proteins known? Is it due to the transient nature of E2-E3 interaction and therefore is the methodology used suitable? This needs to be clarified.

2. "This experimental setup provides a system to monitor PEX protein modulation independently from possible changes in endogenous PEX gene expression (Fig. 4e)." While this is correct, the authors cannot exclude that by overexpressing the substrate, you are not overwhelming the E3 capacity to ubiquitinate the substrate, resulting in stabilisation of Flag-tagged PEX. Therefore, this result would benefit from the confirmation that this is also true for endogenous PEX proteins using specific antibodies. Same for Fig. 6e.

3. "To this purpose, cells were treated with E2combo siRNAs and either with control NT siRNAs or with a combination of siRNAs to knockdown PEX proteins (PEXcombo, i.e. PEX3, PEX11A, PEX11B, and PEX13 siRNAs) to counteract their upregulation in response to E2combo RNAi (Extended Data Fig. 11)." No Extended Data Fig. 11 is present. This needs to be addressed.

4. "In agreement with the hypothesis, the proportion of functional peroxisomes (i.e. PMP70 puncta with GFP-PTS1) increased upon E2combo RNAi but such increase in peroxisomal

protein import was blunted by PEXcombo knockdown (Fig. 5c).". While this is correct, comparing siPEXcombo samples with or without siE2combo shows an equivalent increase as for siNT samples with or without siE2combo. So, the conclusion is not reflecting the results here to me, especially if the siE2combo+siPEXcombo condition is significantly higher than the siNT+siPEXcombo condition. This needs to be clarified.

5. In Fig.5g, statistical tests with multiple comparison need to be performed, as it is not correct to assume that because the siNT+siNT condition is significantly different to siUBA1+siNT but not different to siUBA1+siPEX that siUBA1+siNT is significantly different to siUBA1+siPEX. This needs to be addressed to support the authors' conclusion.

Minor comments:

1. This statement is not accurate: "Mono-ubiquitination regulates protein import, localization, and stability^{1,2,4,5,14-16} whereas poly-ubiquitination promotes degradation via the proteasome or the autophagy lysosome system, depending on the ubiquitin lysine residue that is used to build the poly ubiquitin chains". Monoubiquitination of small proteins can target them to the proteasome and poly-ubiquitination is not only involved in degradation as it also transduces signalling pathways. It may also be worth mentioning that other residues than lysine can be ubiquitinated.

2. In Fig. 1d, it's not clear if the JUMPptm analysis is only based on RPS27A or on the ubiquitination status of the most abundant proteins detected by TMT. This needs to be clarified. In the scenario, it's only based on RPS27A, it would be interesting to include known substrates being modified selectively by other chain types, such as K63.

3. Colour code used in Fig. 1f-h and ED Fig3a need to be specified.

4. "However, these E2s physically interacted with other E2s, which could indicate they act predominantly as an accessory to other E2-E3 ubiquitin chain editing complexes without binding to an E3 directly.". This could also be due to the fact that it's unlikely that all E3s are expressed in HEK293T cells or that the interaction is below the detection threshold. This could be assessed by looking at E3 coverage from their HEK293T proteomic data.

5. "For example, UBE2V1/2 (which did not interact with an E1) was found in complex with UBE2N, which in turn interacted with UBA1 (Fig. 2f).". This is rather counterintuitive, as if UBE2V1/2 interact with UBE2N, UBA1 IP should co-immunoprecipitate the complex UBE2V1/2-UBE2N, unless the interaction is very dynamic or mutually exclusive. This could be discussed in the text.

6. "In addition, also PEX5 was included although its levels are not regulated by UBA1/E2combo knockdown (Fig. 4a-b).". It's not clear why PEX5 has been added, especially that Flag-tagged PEX5 is not localising to peroxisome (ED Fig. 6). This needs to be clarified.

7. "these analyses indicate that UBA1/E2combo RNAi increases the levels of PEX3-FLAG, PEX11A-FLAG, and PEX12-FLAG (Fig. 4e)". PEX11A-FLAG is not significantly increased upon siRNA, unless wrongly annotated. This needs to be changed.

8. This title is not accurate: "Integrated proteome and transcriptome analyses identify protein subsets that are modulated by E2 ubiquitin-conjugating enzymes in a mRNA

independent manner". While transcriptomics show that it is post-transcriptional, it does not give information about mRNA translation, which is an mRNA-dependent mechanism. So, "posttranscriptionally" would be more accurate.

9. In the sentence "by global reduction in E2 function, which was induced by UBA1 RNAi and by combining siRNAs to achieve the concomitant knockdown of 19 E2s ("E2 combo)". It should be "in E1 function".

Reviewer #3 (Remarks to the Author):

I have specifically looked at the lipidomics experiment and the link that is suggested with peroxisomal lipid metabolism.

Major:

Annotation of lipids

The paper that is cited for the lipidomics analysis in the methods (ref 148) presents the JUMPm software package that is used to process LC-MS/MS data, in the paper it concerns metabolomics, not lipidomics data. The reference cited in the results section (ref 88) also concerns metabolomics, not lipidomics. This is not a trivial difference as lipid annotation is very different than small molecule annotation. The description of the method is not sufficient to ascertain the validity of the lipidomics annotation and its results and I have serious doubts about it and the suggested association of UBA1/E2 knockdown with peroxisomal lipid metabolism. Where is the list of the 8070 lipid species detected including the main classes/categories described in the supplemental information in Fig 5f part II? How was annotation done (methods description and references do not mention lipids = not sufficient)? Realizing the annotation of 8070 lipids would be a great accomplishment, which I have not encountered before, this makes me weary or jealous if true. In Fig 5f part I the annotation of MoNA, Lipidblast and "Database name" frequently do not match, which is not encouraging. Also, frequently the annotated features are not lipids (leucine, chloroxine, oxytetracycline). The lipids in this list are not regulated by peroxisomes including those in Fig 5g. The glucuronide of beta-sitosterol / campesterol are exogenous plant sterols not metabolized in peroxisomes. Same goes for the vitamin D3 derivative. The PS plasmalogen is a very very low abundant lipid normally not detected but may have passed the peroxisome if correctly annotated. The "normal" PS is not peroxisomal. PGE1 could be, but not a clear-cut relation with peroxisomal metabolism.

In summary:

- I have serious doubts about the quality of the lipidomics –mainly how annotation was done- and not nearly sufficient (supplementary) data to investigate this.
- The lipidomics data do absolutely not clearly link peroxisomal lipid metabolism and do not support the conclusion that "lipid homeostasis is rewired by UBA1/E2 knockdown, and that this largely depends on PEX protein upregulation and peroxisomal function."

Minor:

Page 10: peroxisomes are not involved in (chole)sterol metabolism, as frequently still wrongly stated. They are involved in bile acid synthesis, which are sterol derivatives but I suggest to change sterol to bile acid.

Reviewer #4 (Remarks to the Author):

This manuscript used quantitative proteomics to identify peroxisome proteins which are regulated by UBA1 or E2s through knocking down experiments. The authors analyzed the data in depth and used biochemical experiments to discover a homeostatic mechanism that preserves organelle function in cells with decreased ubiquitination. Although the manuscript is comprehensive, there are several issues should be taken care of.

- 1. The abstract almost did not describe the methods and the major discoveries of the manuscript. It only described the conclusion of the manuscript. It is better to write in a way that most readers in the ubiquitin community could understand the importance of the work.**
- 2. Figure 4. PEX13 is upregulated in Figure 4a while it is not changed in Figure 4e. PEX12 is downregulated in Figure 4a while it is slightly upregulated in Figure 4e. What are the possible reasons for this discrepancy?**
- 3. Figure 6: It is not easy to understand Figure 6a and 6b. Is it possible to use other ways to describe the data?**
- 4. Figure 6d and Figure 6e. The change of PEX13 and PEX12 are not consistent in these two figures. What are the reasons for these contradictory results?**
- 5. Figure 6: It is hard to understand that PEX13 is increased while its mRNA and ubiquitination is not changed after knocking down UBE2D. How could this happen?**
- 6. In the model of Figure 7k, PEX5 is mono-ubiquitinated. But the Western blotting in Figure 6f clearly showed that it is polyubiquitinated. It seems that the data do not support the model. Is there any reason for this?**
- 7. The authors knocked down UBA1 or combined E2s. Are there any physiological or pathological conditions that correspond to this situation?**
- 8. How do the authors determine which of 19 E2s to knock down in the combo knockdown experiments?**
- 9. Most of the ubiquitination images in Extended Data Figure 7 have two heavy bands. These two bands should be noted in the figure legend. Are they the heavy and light chains of the antibody used for IP? In addition, some of the lanes in e and f do not have these two bands.**
- 10. $\text{Log}_2\text{FC} < -0.2$, $\text{Log}_{10}(\text{P-value})$, 2 and 10 should be subscript in figure and text.**
- 11. The following sentence should be slightly modified since the proteomic data have been deposited. "The proteomics data underlying the study will be deposited in the Proteomics Identification Database (PRIDE) database upon publication."**

RESPONSE TO THE REVIEWERS' COMMENTS

Reviewer #1 (Remarks to the Author):

Comments to the Author:

In this manuscript, Hunt and colleagues utilized quantitative proteomics and functional genomics to characterize the perturbation of cellular proteome upon decreased ubiquitination. The authors found that the knockdown of UBA1 ubiquitin-activating enzyme and E2 ubiquitin-conjugating enzymes reshapes the proteome, and integrated analyses reveal specific E2 target proteins. Intriguingly, the knockdown of either UBA1 or E2 enzymes increases peroxins and peroxisomal import function. Thus, the work uncovers a novel interplay between the protein ubiquitination system and peroxisomal import function. I have a few concerns as listed below.

Thank you for finding our work novel and intriguing. We have addressed all the specific comments by revising the manuscript and the figures accordingly, as explained here below.

1. In Figure 4, the authors show that UBA1/E2combo RNAi increases the levels of PEX3, PEX11A and PEX12. Except for PEX12, PEX3, and PEX11A are not directly involved in the regulation of peroxisomal import. PEX3 mainly works with PEX19 in controlling peroxisome de novo biogenesis, while PEX11 regulates peroxisome fission. The increased levels of PEX3 and PEX11A might result in a higher number of peroxisomes. The authors should check this possibility by quantifying the number of PMP70-positive puncta in UBA1/E2combo knockdown cells.

We have now examined the average number of peroxisomes (PMP70-positive puncta) per cell and found no significant increase when comparing UBA1/E2combo knockdown to control NT siRNA treated cells. This new data is shown in Supplementary Fig. 11.

PEX14 antibody should also be used to validate PMP70 immunostaining results.

Supplementary Figure 8c now indicates that PMP70 staining largely overlaps with PEX14 immunostaining.

In Figure 5d, the knockdown of either PEX3 or PEX11 did not significantly block UBA1/E2combo KD-induced peroxisomal import. This is likely because the two proteins are not directly involved in peroxisome import. But the authors may examine the requirement of PEX3 or PEX11 for UBA1/E2-regulated peroxisome de novo biogenesis.

We have now examined epistasis between UBA1/E2combo knockdown and PEX3 and PEX11 but found no significant effect in regulating the number of peroxisomes (PMP70-positive puncta). This new data is shown in Supplementary Fig. 11.

2. In Figure 5f-5g, the authors performed lipidomics and identified ~100 lipid species that are modulated by UBA1 and E2 in a peroxin-dependent manner. However, it is unclear whether the changes in lipid profiles are due to enhanced peroxisome function or due to altered metabolism in other organelles like ER and mitochondria. Because it is known that peroxisomes interact with many organelles to maintain cellular homeostasis, it would be nice to show what lipid species are directly linked to peroxisome-specific metabolism (like VLCFA beta-oxidation), and which lipids are related to ER or mitochondrial metabolism.

We have now performed a comprehensive analysis of the lipid metabolic pathways that are modulated by UBA1/E2combo RNAi (Fig. 6c-g). Such analysis of lipid metabolic pathways was determined by using the BioPAN software, available at lipidmaps.org/biopan (ref. 96-97 cited in the manuscript).

These analyses revealed several pathway-level alterations in lipid metabolism in response to UBA1 and E2combo knockdown (Fig. 6c-f), including the promotion of plasmalogen biosynthesis (Fig. 6g), which is known to critically depend on peroxisomes. Specifically, siRNAs for UBA1 and for E2combo promote the conversion of PE-O into PE-plasmalogen (P-PE) and PC-plasmalogen (P-PC), and this is impeded by concomitant PEX knockdown (siUBA1/siE2combo +siPEX) compared to control NT siRNAs (siUBA1/siE2combo +siNT) (Fig. 6g). Altogether, these results indicate that lipid homeostasis is rewired by UBA1/E2 knockdown in a largely PEX-dependent manner.

We acknowledge that lipid changes induced by UBA1/E2combo RNAi are likely dependent on multiple organelles apart peroxisomes. However, we are not aware of any automated way to assign a lipid species to a specific organelle as in many cases lipid metabolism occurs across multiple organelles.

We have amended the Results section of the manuscript accordingly to indicate that the lipidomic changes induced by UBA1/E2combo RNAi also included lipid species that may not be directly modulated by peroxisomes but rather by other organelles, such as mitochondria, which are remodeled at the proteome level by UBA1/E2combo RNAi (Fig. 3f-i) and which are also known to interact with peroxisomes via membrane contact sites (ref. 93-95 cited in the manuscript).

3. In Figure 6, I wonder if the regulation of PEX ubiquitination by UBE2D1/2/3 is direct. Can authors perform a Co-IP analysis to examine the possible interaction between UBE2D1/2/3 and individual peroxin? **We have performed immunoprecipitation experiments with baits consisting of 8 E1s and 28 E2s that are expressed in HEK293T cells, including UBE2D1, UBE2D2, and UBE2D3. In these immunoprecipitation experiments (which resulted in the E1-E2 interactome reported in Fig.2), no peroxins were detected, presumably because the interaction between UBE2Ds (and other E2s) and PEX is transient and mediated by an E3 ubiquitin ligase. However, UBE2D1 has been previously reported to interact with PEX10 in a “reconstituted complex” (PMID: 18644345, <https://pubmed.ncbi.nlm.nih.gov/18644345/>).**

In Figure 6d, I guess that “UBE2D knockdown” was missed from the figure panel labels.

The previous Fig. 6d is now Fig. 7d. The x-axis of Fig.7d now indicates that the data refers to “UBE2D1/2/3 siRNAs vs NT siRNAs”.

4. In Figure 6f (and the model Figure 7k), the authors indicate that PEX5 ubiquitination is downregulated by UBE2D KD. However, it is unclear whether both mono-ubiquitination and poly-ubiquitination of PEX5 are reduced. If mono-ubiquitination of PEX5 is reduced, one would expect to find impaired peroxisomal import. The authors need carefully examine mono-ubiquitination vs. poly-ubiquitination of PEX5 upon UBE2D KD.

The previous Fig. 6f and 7k are now Fig. 7f and 9a, respectively. Figure 7f demonstrates that UBE2D siRNAs reduce PEX5 ubiquitination. Our analysis with anti-ubiquitin antibodies does not distinguish between mono-ubiquitinated versus poly-ubiquitinated PEX5, presumably because the predicted band of mono-ubiquitinated PEX5 (~85 kDa) is weak compared to the higher bands of poly-ubiquitinated PEX5 in Fig. 7f. Therefore, we have amended the manuscript and made sure that we never refer to mono-ubiquitination when discussing our data on PEX5 ubiquitination. However, we would still like to mention previous studies that have demonstrated the importance of PEX5 mono-ubiquitination for peroxisomal protein import (ref. 47-50 cited in the manuscript).

5. In Figure 7, I suggest the authors take a look at the peroxisome number in fly muscle as well, for the same reason mentioned above.

We have used an antibody raised against Drosophila PMP70 and previously used for detecting peroxisomes in Drosophila (PMID: 32523050). However, the staining that we obtained from skeletal muscle was diffuse and aspecific and therefore we are afraid we cannot use this tool to detect the total number of peroxisomes in Drosophila muscles. Nonetheless, our analyses in Fig. 8 have found a higher number of YFP-PTS1 positive puncta (i.e. peroxisomes with functional import) in the muscles with Uba1 RNAi and eff/UBE2D RNAi compared to controls (white RNAi).

Reviewer #2 (Remarks to the Author):

Protein ubiquitination is one of the major post-translational modifications in cells regulating most, if not all, cellular processes. Ubiquitination has various outcomes including modulation of protein stability, localization, function, and degradation. How cells respond to an overall inhibition of ubiquitinylation remains elusive. In this manuscript the authors explore the impact of UBA1/E2s knockdown on the cellular proteome using quantitative proteomics. They found that specific compartments are more affected by a moderate, partial reduction in ubiquitin conjugation, such as peroxisome, ER and mitochondria. They further define a network of E2 cross-interaction via IP-MS interactome mapping. Having found a more profound effect of UBA1/E2s knockdown on peroxisomal proteins, they embarked on defining the underlying mechanisms. They identified that UBA1/E2 knockdown boosts peroxisomal import via the counterbalancing upregulation of peroxins necessary for PEX5 docking to the peroxisomal membrane using both HEK293T cells and drosophila. They argue that homeostatic mechanisms are triggered to preserve organelle homeostasis upon reduction of overall ubiquitinylation.

Better understanding how cells adapt to ubiquitin conjugation defect along with its impact on proteostasis is an important and timely contribution. Data presented in the manuscript support overall authors' conclusion, although some specific conclusions are less convincing (see below). On the whole, the paper is well written and interesting but may have benefited from being more focused, as some data don't bring much to the story (e.g., IP-MS interactome mapping found only few E3s co-immunoprecipitated with E2s and those E3s were not further studied in the paper). In summary, I do think that this paper will be a good addition to the ubiquitin and proteostasis field and I would support publication of this manuscript providing the more specific concerns listed below are addressed.

Thank you for finding our work important and timely. We have addressed all the comments as explained in detail here below.

Major comments:

1. Fig. 2f shows the overall E2 interactome, do the authors know why only ~40 E3/DUBs are identified out of the ~620-800 E3/DUB proteins known? Is it due to the transient nature of E2-E3 interaction and therefore is the methodology used suitable? This needs to be clarified.

Supplementary Figure 5 compares the average spectral counts obtained by TMT-MS for all E3s detected in HEK293s versus those detected in association with E2s (IP-MS detected). Those detected by IP-MS were typically more abundant and therefore this is likely the reason they were relatively

few E3s detected by IP-MS: there might indeed be a detection threshold based on their relative abundance in HEK293 cells as well as because of the transient nature of E2-E3 interactions. While other labeling methods may be useful to overcome transient interactions, they also have limitations due to the high size of the tag added to the bait protein and the harsh chaotropic elution (compared to the mild FLAG-peptide competitive elution) which can lead to high levels of noise. Nonetheless, our interactome has defined E1-E2 cross-interactions and, although with limited coverage, has also uncovered some E2-E3 interactions.

2. "This experimental setup provides a system to monitor PEX protein modulation independently from possible changes in endogenous PEX gene expression (Fig. 4e)". While this is correct, the authors cannot exclude that by overexpressing the substrate, you are not overwhelming the E3 capacity to ubiquitinate the substrate, resulting in stabilisation of Flag-tagged PEX. Therefore, this result would benefit from the confirmation that this is also true for endogenous PEX proteins using specific antibodies. Same for Fig. 6e. **We agree and have tried a few commercial antibodies to detect endogenous PEX proteins by western blots. However, we have found these antibodies to be unreliable (i.e. aspecific). Nonetheless, our TMT mass spectrometry studies have detected endogenous PEX proteins (Fig. 4a-b and Fig. 7a-d). Moreover, our analysis of FLAG-tagged PEX proteins may have the limitation of overwhelming the system but still provides evidence for PEX protein modulation by the proteasome (Fig. 4f).**

3. "To this purpose, cells were treated with E2combo siRNAs and either with control NT siRNAs or with a combination of siRNAs to knockdown PEX proteins (PEXcombo, i.e. PEX3, PEX11A, PEX11B, and PEX13 siRNAs) to counteract their upregulation in response to E2combo RNAi (Extended Data Fig. 11)". No Extended Data Fig. 11 is present. This needs to be addressed.

We have amended the Supplementary Information, which now includes 13 Supplementary Figures.

4. "In agreement with the hypothesis, the proportion of functional peroxisomes (i.e. PMP70 puncta with GFP-PTS1) increased upon E2combo RNAi but such increase in peroxisomal protein import was blunted by PEXcombo knockdown (Fig. 5c)". While this is correct, comparing siPEXcombo samples with or without siE2combo shows an equivalent increase as for siNT samples with or without siE2combo. So, the conclusion is not reflecting the results here to me, especially if the siE2combo+siPEXcombo condition is significantly higher than the siNT+siPEXcombo condition. This needs to be clarified.

This is an RNAi treatment to reduce the mRNA levels of the target transcripts. Therefore, it does not reflect a complete loss of function, and the decreased level of remaining PEX proteins are still subject to regulation by E2s.

5. In Fig.5g, statistical tests with multiple comparison need to be performed, as it is not correct to assume that because the siNT+siNT condition is significantly different to siUBA1+siNT but not different to siUBA1+siPEX that siUBA1+siNT is significantly different to siUBA1+siPEX. This needs to be addressed to support the authors' conclusion.

We have removed the examples of modulated lipids originally shown in Fig. 5g and substituted this with Fig. 6c-g which displays the BioPAN analysis of lipid metabolic pathways modulated by UBA1/E2combo siRNAs.

Supplementary Table 5 reports all the lipidomic data and the statistical cross-comparison of all the experimental groups.

Minor comments:

1. This statement is not accurate: "Mono-ubiquitination regulates protein import, localization, and stability^{1,2,4,5,14-16} whereas poly-ubiquitination promotes degradation via the proteasome or the autophagy lysosome system, depending on the ubiquitin lysine residue that is used to build the poly ubiquitin chains". Monoubiquitination of small proteins can target them to the proteasome and poly-ubiquitination is not only involved in degradation as it also transduces signalling pathways. It may also be worth mentioning that other residues than lysine can be ubiquitinated.

We have amended the Introduction accordingly: "E2 recruits a client E3, which in turn acts as an enzyme or as a scaffold to link ubiquitin typically to a lysine (but also to other residues) on a specific target protein" and "Mono- and poly-ubiquitination constitute a complex code that regulates protein localization, function, and degradation".

2. In Fig. 1d, it's not clear if the JUMPptm analysis is only based on RPS27A or on the ubiquitination status of the most abundant proteins detected by TMT. This needs to be clarified. In the scenario, it's only based on RPS27A, it would be interesting to include known substrates being modified selectively by other chain types, such as K63.

The JUMPptm analysis shown in Fig.1d is based only on RPS27A (ubiquitin) because the specific linkage identified by JUMPptm is on the ubiquitin protein (i.e. K11/48/63 etc.) and it is, therefore, the only way that poly-ubiquitin linkages could be measured from this analysis (they represent the entirety of poly-ubiquitin regardless of the substrates they are attached to). Examining other substrates with differential ubiquitination would only indicate which lysine or other residue of the substrate was ubiquitinated but not what type of poly-ubiquitin linkage is present.

We are now also including Supplementary Fig. 3 and Supplementary Table 2 in support of the JUMPptm analysis shown in Fig.1d.

3. Colour code used in Fig. 1f-h and ED Fig3a need to be specified.

Thank you for pointing this out, this has been added.

4. "However, these E2s physically interacted with other E2s, which could indicate they act predominantly as an accessory to other E2-E3 ubiquitin chain editing complexes without binding to an E3 directly.". This could also be due to the fact that it's unlikely that all E3s are expressed in HEK293T cells or that the interaction is below the detection threshold. This could be assessed by looking at E3 coverage from their HEK293T proteomic data.

Supplementary Fig. 5a compares the average spectral counts for all proteins detected by TMT in HEK293T cells versus the proteins detected by IP-MS in association with E1/E2s. Those detected by IP-MS are typically more abundant (Supplementary Fig. 5a-b). Likewise, E3 ubiquitin ligases and deubiquitinating enzymes retrieved in association with E2s in IP-MS experiments are on average more abundant than E3s/DUBs that are detected by TMT but not by IP-MS experiments (Supplementary Fig. 5c-d). Altogether, these new analyses suggest that there is a threshold that precludes the detection of poorly-abundant E3 ligases in HEK293 cells, in addition to the transient nature of E2-E3 interactions.

5. "For example, UBE2V1/2 (which did not interact with an E1) was found in complex with UBE2N, which in turn interacted with UBA1 (Fig. 2f)". This is rather counterintuitive, as if UBE2V1/2 interact with UBE2N, UBA1

IP should co-immunoprecipitate the complex UBE2V1/2-UBE2N, unless the interaction is very dynamic or mutually exclusive. This could be discussed in the text.

We have added the following sentences to the Results section: “Although this would suggest that UBE2V1/2 should co-immunoprecipitate with a UBA1-UBE2N complex, this did not occur, presumably because these interactions are extremely dynamic or mutually exclusive. Moreover, if the interaction of UBE2V1/2 to UBA1 is indirect and occurs via UBE2N, only a small fraction of UBE2V1/2 may associate with the UBA1 bait and thus be missed because below the MS detection threshold”.

Direct interactions with a bait protein are more readily detected than those that are indirect (i.e. that have a greater degree of separation by other interacting proteins). For example, Protein 1 interacts with Protein 2. Protein 2 interacts with Protein 3. If Protein 1 is the bait and 10% of all of Protein 2 is detected in association with Protein 1 and this is within the MS limits, then Protein 2 will be detected as interacting with Protein 1. If the same logic follows and 10% of all Protein 3 can be detected in association with Protein 2 as bait and is within the MS limits, then Protein 3 will be detected as interacting with Protein 2. However, on this basis, you would anticipate a much smaller amount of Protein 3 to interact with the Protein 1 bait, due to the additional degree of separation via Protein 2: 10% of Protein 2 interacts with Protein 1 and 10% of Protein 3 interacts with the 10% of Protein 2, leading to around a total of 1% of Protein 3 interacting with Protein 1, which probably does not reach the MS detection limits.

6. “In addition, also PEX5 was included although its levels are not regulated by UBA1/E2combo knockdown (Fig. 4a-b)”. It’s not clear why PEX5 has been added, especially that Flag-tagged PEX5 is not localising to peroxisome (ED Fig. 6). This needs to be clarified.

We have now specified the logic of PEX5 inclusion. Although PEX5 protein levels are not modulated by UBA1/E2 knockdown, we have included PEX5 in these analyses because of its importance in peroxisomal protein import, for which it acts as a cargo receptor that cycles from the peroxisomal membrane to the cytol, a process that has been previously found to be regulated by PEX5 mono-ubiquitination (ref. 47-50 cited in the manuscript).

7. “these analyses indicate that UBA1/E2combo RNAi increases the levels of PEX3-FLAG, PEX11A-FLAG, and PEX12-FLAG (Fig. 4e)”. PEX11A-FLAG is not significantly increased upon siRNA, unless wrongly annotated. This needs to be changed.

This has been corrected.

8. This title is not accurate: “Integrated proteome and transcriptome analyses identify protein subsets that are modulated by E2 ubiquitin-conjugating enzymes in a mRNA independent manner”. While transcriptomics show that it is post-transcriptional, it does not give information about mRNA translation, which is an mRNA-dependent mechanism. So, “posttranscriptionally” would be more accurate.

This has been corrected.

9. In the sentence “by global reduction in E2 function, which was induced by UBA1 RNAi and by combining siRNAs to achieve the concomitant knockdown of 19 E2s (“E2 combo”)”. It should be “in E1 function”.

This has been corrected.

Reviewer #3 (Remarks to the Author):

I have specifically looked at the lipidomics experiment and the link that is suggested with peroxisomal lipid metabolism.

Major:

Annotation of lipids

The paper that is cited for the lipidomics analysis in the methods (ref 148) presents the JUMPm software package that is used to process LC-MS/MS data, in the paper it concerns metabolomics, not lipidomics data. The reference cited in the results section (ref 88) also concerns metabolomics, not lipidomics. This is not a trivial difference as lipid annotation is very different than small molecule annotation. The description of the method is not sufficient to ascertain the validity of the lipidomics annotation and its results and I have serious doubts about it and the suggested association of UBA1/E2 knockdown with peroxisomal lipid metabolism. How was annotation done (methods description and references do not mention lipids = not sufficient)?

We appreciate the reviewer for bringing up this crucial concern regarding the difference between metabolomics and lipidomics data processing. We acknowledge that the JUMPm software package (current ref. 157 cited in the manuscript) was initially designed for metabolomics data processing, encompassing four major components: data preprocessing, feature peak extraction for individual sample, peak alignment across samples, and metabolite identification for aligned feature peaks. In our study, although we employed the JUMPm for processing our data, we utilized several lipid libraries/databases for lipid identification including an in-house lipid library, the downloaded experimental MoNA spectral library, and the widely used Lipidblast and LipidMAP libraries. Moreover, we want to emphasize that the scoring algorithm we adopted in the JUMPm program for lipid identification is the widely used dot product method. To clarify this point, we have amended accordingly the method section of the manuscript.

Realizing the annotation of 8071 lipids would be a great accomplishment, which I have not encountered before, this makes me weary or jealous if true. In Fig 5f part I the annotation of MoNA, Lipidblast and "Database name" frequently do not match, which is not encouraging. Also, frequently the annotated features are not lipids (leucine, chloroxine, oxytetracycline).

We acknowledge that also metabolites are identified in these analyses and therefore we have modified the text accordingly to refer to "lipidome/metabolome profiling" rather than simply to "lipidome profiling".

Regarding the number of lipids and metabolites (i.e., 8,071) identified in this study, this count is relatively large because it includes both lipids and metabolites and because we searched our MS data using four distinct libraries/databases, including our in-house library, the MassBank of North America (MoNA) library, LipidBlast, and the LipidMAP database.

Several reasons contribute to the discrepancies observed in annotations across various libraries: (i) the same feature was annotated with different names across different libraries, albeit falling within the same lipid group due to their similar structures; (ii) certain lipids are exclusive to a specific library, whereas they are absent from other libraries. Consequently, annotation is assigned in the relevant library, but incorrect nomenclature was used in alternative libraries; (iii) while the same lipid was identified, discrepancies arise due to differing terminologies employed by different library sources.

Where is the list of the 8071 lipid species detected including the main classes/categories described in the supplemental information in Fig 5f part II?

The original lipidomics/metabolomics data (8071 lipid/metabolite species) is provided in Supplementary Table 5. The subset of lipids shown in Fig. 5f part II (currently Fig. 6b) is included in the Source Data file (in the “Fig6b – part1” and “Fig6b – part 2” tabs).

The lipids in this list are not regulated by peroxisomes including those in Fig 5g. The glucuronide of beta-sitosterol / campesterol are exogenous plant sterols not metabolized in peroxisomes. Same goes for the vitamin D3 derivative. The PS plasmalogen is a very very low abundant lipid normally not detected but may have passed the peroxisome if correctly annotated. The “normal” PS is not peroxisomal. PGE1 could be, but not a clear-cut relation with peroxisomal metabolism.

We have removed the previous Fig. 5g and substituted these graphs reporting a few examples of modulated lipids with a comprehensive analysis of the lipid metabolic pathways that are modulated by UBA1/E2combo RNAi (Fig. 6c-g). Such analysis of lipid metabolic pathways was done with the BioPAN software, available at lipidmaps.org/biopan (ref. 96-97 cited in the manuscript).

These analyses revealed several pathway-level alterations in lipid metabolism in response to UBA1 and E2combo knockdown (Fig. 6c-f), including the promotion of plasmalogen biosynthesis (Fig. 6g), which is known to critically depend on peroxisomes. Specifically, siRNAs for UBA1 and for E2combo promote the conversion of PE-O into PE-plasmalogen (P-PE) and PC-plasmalogen (P-PC), and this is impeded by concomitant PEX knockdown (siUBA1/siE2combo +siPEX) compared to control NT siRNAs (siUBA1/siE2combo +siNT) (Fig. 6g). Altogether, these results indicate that lipid homeostasis is rewired by UBA1/E2 knockdown in a largely PEX-dependent manner.

In summary:

- I have serious doubts about the quality of the lipidomics –mainly how annotation was done- and not nearly sufficient (supplementary) data to investigate this.

We have now improved the methods section. All the lipidomics/metabolomics data is provided in Supplementary Table 5 and in the Source Data file.

- The lipidomics data do absolutely not clearly link peroxisomal lipid metabolism and do not support the conclusion that “lipid homeostasis is rewired by UBA1/E2 knockdown, and that this largely depends on PEX protein upregulation and peroxisomal function.”

We have now utilized the BioPAN software to analyze the lipidomics data and obtain a comprehensive view of how lipid metabolic pathways are re-wired by a reduction in cellular ubiquitination capacity. In particular, these analyses reveal several pathway-level alterations in lipid metabolism in response to UBA1 and E2combo knockdown (Fig. 6c-f), including the promotion of plasmalogen biosynthesis (Fig. 6g), which is known to critically depend on peroxisomes. Specifically, siRNAs for UBA1 and for E2combo promote the conversion of PE-O into PE-plasmalogen (P-PE) and PC-plasmalogen (P-PC), and this is impeded by concomitant PEX knockdown (siUBA1/siE2combo +siPEX) compared to control NT siRNAs (siUBA1/siE2combo +siNT) (Fig. 6g). Altogether, these results indicate that lipid homeostasis is rewired by UBA1/E2 knockdown in a largely PEX-dependent manner.

Minor:

Page 10: peroxisomes are not involved in (chole)sterol metabolism, as frequently still wrongly stated. They are involved in bile acid synthesis, which are sterol derivatives but I suggest to change sterol to bile acid.

We have now removed any mention of sterol metabolism being connected with peroxisomes. This previous statement was primarily based on this paper:

Chu, B. B. *et al.* Cholesterol transport through lysosome-peroxisome membrane contacts. *Cell* 161, 291-306 (2015). <https://doi.org:10.1016/j.cell.2015.02.019>

Reviewer #4 (Remarks to the Author):

This manuscript used quantitative proteomics to identify peroxisome proteins which are regulated by UBA1 or E2s through knocking down experiments. The authors analyzed the data in depth and used biochemical experiments to discover a homeostatic mechanism that preserves organelle function in cells with decreased ubiquitination. Although the manuscript is comprehensive, there are several issues should be taken care of.

1. The abstract almost did not describe the methods and the major discoveries of the manuscript. It only described the conclusion of the manuscript. It is better to write in a way that most readers in the ubiquitin community could understand the importance of the work.

Thank you for pinpointing the importance of our work for the ubiquitin community. We have now revised the abstract accordingly to convey that, in addition to reporting a novel adaptive response to decreased ubiquitination, our work provides an important resource for the ubiquitin community by defining the E1-E2 interactome and the E2 specificity in modulating target protein levels in human cells.

2. Figure 4. PEX13 is upregulated in Figure 4a while it is not changed in Figure 4e. PEX12 is downregulated in Figure 4a while it is slightly upregulated in Figure 4e. What are the possible reasons for this discrepancy?

This discrepancy may arise from the fact that endogenous PEX12 and PEX13 are detected in Fig. 4a-b whereas exogenously expressed, Flag-tagged PEX12 and PEX13 are monitored in Fig. 4e.

3. Figure 6: It is not easy to understand Figure 6a and 6b. Is it possible to use other ways to describe the data?

We have now made a new supplementary figure (Supplementary Fig. 12) to display this data in another way.

4. Figure 6d and Figure 6e. The change of PEX13 and PEX12 are not consistent in these two figures. What are the reasons for these contradictory results?

Previous Fig. 6d-e are now Fig. 7d-e. This discrepancy may arise from the fact that endogenous PEX12 and PEX13 are detected in Fig. 7d whereas exogenously expressed, Flag-tagged PEX12 and PEX13 are monitored in Fig. 7e.

5. Figure 6: It is hard to understand that PEX13 is increased while its mRNA and ubiquitination is not changed after knocking down UBE2D. How could this happen?

Previous Fig. 6 is now Fig. 7. We would like to point out that PEX13 ubiquitination is indeed reduced by UBE2D knockdown, as shown in Fig. 7f. Our data, therefore, suggest that UBE2D regulates PEX13 protein levels via ubiquitination and proteasome-mediated degradation.

6. In the model of Figure 7k, PEX5 is mono-ubiquitinated. But the Western blotting in Figure 6f clearly showed that it is polyubiquitinated. It seems that the data do not support the model. Is there any reason for this?

The previous Fig. 6f and 7k are now Fig. 7f and 9a, respectively. Figure 7f demonstrates that UBE2D siRNAs reduce PEX5 ubiquitination. Our analysis with anti-ubiquitin antibodies does not distinguish between mono-ubiquitinated versus poly-ubiquitinated PEX5, presumably because the predicted band of mono-ubiquitinated PEX5 (~85 kDa) is weak compared to the higher bands of poly-ubiquitinated PEX5 in Fig. 7f. Therefore, we have amended the manuscript and made sure that we never refer to mono-ubiquitination when discussing our data on PEX5 ubiquitination. However, we would still like to mention previous studies that have demonstrated the importance of PEX5 mono-ubiquitination for peroxisomal protein import (ref. 47-50 cited in the manuscript).

7. The authors knocked down UBA1 or combined E2s. Are there any physiological or pathological conditions that correspond to this situation?

Decreased activity of UBA1 (and hence decreased E2-mediated ubiquitin conjugation) is the cause of VEXAS syndrome, an adult-onset systemic inflammatory condition that leads to progressive bone marrow failure because of defects in the function of hematopoietic stem cells (ref. 30-34 in the manuscript). Moreover, missense mutations that reduce UBA1 activity cause infantile spinal muscular atrophy X-linked 2 (SMAX2), a disease characterized by muscle weakness (ref. 35-40 in the manuscript). In addition to VEXAS and SMAX2, it has also been proposed that decreased activity of UBA1 and E2s may also contribute to the etiology of other diseases (ref. 41-46 in the manuscript).

8. How do the authors determine which of 19 E2s to knock down in the combo knockdown experiments?

The siRNAs for the 32 E2s expressed by HEK293 cells were combined to obtain an E2combo pool of siRNAs, which however significantly reduced the expression of 19 out of the 32 siRNA-targeted E2s (others may have been reduced but not significantly).

9. Most of the ubiquitination images in Extended Data Figure 7 have two heavy bands. These two bands should be noted in the figure legend. Are they the heavy and light chains of the antibody used for IP? In addition, some of the lanes in e and f do not have these two bands.

These are most likely mouse IgG heavy and light chains around 50 and 25 kDa from the IP reacting with the secondary antibody. They may derive from small amounts of resin with anti-Flag IgG that gets loaded on the gel (some lanes may lack these bands because no resin was loaded in that case). Their presence does interfere with the measurement of poly-ubiquitin and Flag-tagged proteins. The figure legend of Supplementary Fig. 9 (previous Extended Data Fig. 7) has been revised accordingly.

10. $\text{Log}_2\text{FC} < -0.2$, $\text{Log}_{10}(\text{P-value})$, 2 and 10 should be subscript in figure and text.

This has been changed in both the figures and text.

11. The following sentence should be slightly modified since the proteomic data have been deposited. "The proteomics data underlying the study will be deposited in the Proteomics Identification Database (PRIDE) database upon publication."

Thank you for pointing this out. All the data has been deposited and therefore we have revised this sentence accordingly.

REVIEWER COMMENTS

Reviewer #1 (Remarks to the Author):

The authors have fully addressed my previous comments. I recommend it for publication in Nature Communications.

Reviewer #2 (Remarks to the Author):

I thank the authors for their detailed response to questions. I think that the authors address my concerns in a satisfactory manner, and that their data now better support their conclusions. Limitations are more clearly stated and the part with statistical issue has been replaced. To this end, I have no further concerns and support publication.

Reviewer #3 (Remarks to the Author):

I would like to thank the authors for their detailed response and changes they have made to improve the manuscript. Again I have concentrated on the metabolomics/lipidomics analysis as this is part of the functional validation of the findings at the level of ubiquination that I cannot evaluate.

Concerning the annotation of metabolites/lipid species. The authors acknowledge the fact that the use of multiple metabolite libraries cause multiple annotation of the same molecule thus resulting in an unrealistic and incorrect total number of annotated metabolites. While I now understand how the problem is caused, I do not see changes in the manuscript to address this point as still it is presented as if 8071 metabolites were identified. This is not the case and should be changed.

I carefully inspected the 428755_1_data_set_8048546_s09g5b.xlsx file, which is a big improvement over the old metabolomics data file, yet also contains errors (shifted columns/rows). The use of multiple libraries for different compounds is no problem, but data curation should be performed to ascertain the annotation certainty of the features and those with a high enough certainty score should be selected for further analysis in the cell biological context. There lies the fundamental problem:

1. Metabolite identification: Names are used for the identification of the metabolites making it difficult to see how well the three libraries match regarding annotation. To facilitate this the use of HMDB codes, InChI keys or SMILES would be an option. InChI keys and SMILES are present in the separate database tabs but are not in the combined annotated list. Combining this to match annotations should be done to ascertain concordance of annotation between databases. Yet, when eyeballing the data there is very little overlap between the different database annotations, which is a concern. [in tab pos database, titles are shifted btw.]

2. Feature annotation: Looking closer at the data, this does not make a good impression at all. Why were MS2 scores not used to filter out correctly annotated features? With the information at hand, I can only conclude that for most of the metabolites identified by metabolomics identification was performed solely at the MS1 level, which is not acceptable.

Retention time differences between in-house validated metabolites (?) are very large (minutes!) which generates serious doubt about the annotation accuracy and certainty. These features cannot be included as identified metabolites and should only be considered as features in an untargeted fashion, which obviously is not the aim of this analysis. For instance: if I take the 0,5-1,0 MS2 scored metabolites into account of the Pos All tab in 428755_1_data_set_8048546_s09g5b.xlsx:

o MS2>0,5: Library; 51, Mona; 271, LipidMaps; 356. Total: 678 metabolites, not 8071.

3. Another reason for the many "identifications" is that the used LipidBlast database contains many isomers that are definitely not separated on the analytical system (isomers etc.). for example: HexCer-NDS d42:1 is found at two retention times where one is galactosylceramide and the other glucosylceramide with 10 min retention time difference. This is impossible as their retention time will be almost identical.

4. LPE 22:6 is annotated 5 times with retention times ranging from 14-34 min. Which one is correct (no MS2 score for either of them)? This is seen repeatedly in the dataset which further undermines my confidence in the data and the conclusions drawn from it.

5. I also have serious doubt about the ability of the lipidomics analysis to separate alkyl and alkenyl ether lipid species. Looking at the Pos_Compare_Total tab for example I see two features annotated as PC 38:5e, one as PC(O-18:1(9Z)/20:4(8Z,11Z,14Z,17Z)) and the other as PC(P-18:0/20:4(5Z,8Z,11Z,14Z)), thus the -O and -P isomers with the same exact mass. The retention time difference is 6 min. In our reversed phase analysis this difference is about 0,15 min. This annotation therefore cannot be correct.

As the O- vs. P- difference is an important finding in the lipidomics results which lead the authors to claim that siUBA1/siPEX influences this conversion. If annotation of these species is incorrect, which I consider likely for many of the annotations, the biological consequences inferred from them are too.

6. The authors state "A core set of 100 lipids and metabolites that are significantly and consistently modulated by both UBA1 and E2combo RNAi in a PEX-dependent manner included lipid species known to be regulated by peroxisomes". I could not find this list of lipids. These should be provided to allow evaluation of their peroxisomal nature.

7. In Pos_Compare_Total and Pos_All "Database name" and "LipidBlast name" do not always match (shifted?), this should be corrected.

I therefore strongly recommend to remove the metabolomics experiment as it has serious flaws that were not addressed in the revision. The conclusions drawn from them therefore, unfortunately, cannot be used as the data does not support it.

Reviewer #4 (Remarks to the Author):

The revised manuscript has addressed my comments. It is suitable for publication.

REVIEWER COMMENTS

Reviewer #1 (Remarks to the Author):

The authors have fully addressed my previous comments. I recommend it for publication in Nature Communications.

Response: Thank you for the positive evaluation of our work.

Reviewer #2 (Remarks to the Author):

I thank the authors for their detailed response to questions. I think that the authors address my concerns in a satisfactory manner, and that their data now better support their conclusions. Limitations are more clearly stated and the part with statistical issue has been replaced. To this end, I have no further concerns and support publication.

Response: Thank you for supporting the publication of our study.

Reviewer #3 (Remarks to the Author):

I would like to thank the authors for their detailed response and changes they have made to improve the manuscript. Again I have concentrated on the metabolomics/lipidomics analysis as this is part of the functional validation of the findings at the level of ubiquination that I cannot evaluate.

Concerning the annotation of metabolites/lipid species. The authors acknowledge the fact that the use of multiple metabolite libraries cause multiple annotation of the same molecule thus resulting in an unrealistic and incorrect total number of annotated metabolites. While I now understand how the problem is caused, I do not see changes in the manuscript to address this point as still it is presented as if 8071 metabolites were identified. This is not the case and should be changed.

I carefully inspected the 428755_1_data_set_8048546_s09g5b.xlsx file, which is a big improvement over the old metabolomics data file, yet also contains errors (shifted columns/rows). The use of multiple libraries for different compounds is no problem, but data curation should be performed to ascertain the annotation certainty of the features and those with a high enough certainty score should be selected for further analysis in the cell biological context. There lies the fundamental problem:

1. Metabolite identification: Names are used for the identification of the metabolites making it difficult to see how well the three libraries match regarding annotation. To facilitate this the use of HMDB codes, InChI keys or SMILES would be an option. InChI keys and SMILES are present in the separate database tabs but are not in the combined annotated list. Combining this to match annotations should be done to ascertain concordance of annotation between databases. Yet, when eyeballing the data there is very little overlap between the different database annotations, which is a concern. [in tab pos database, titles are shifted btw.]

2. Feature annotation: Looking closer at the data, this does not make a good impression at all. Why were MS2 scores not used to filter out correctly annotated features? With the information at hand, I can only conclude that for most of the metabolites identified by metabolomics identification was performed solely at the MS1 level, which is not acceptable. Retention time differences between in-house validated metabolites (?) are very large (minutes!) which generates serious doubt about the annotation accuracy and certainty. These features cannot be included as identified metabolites and should only be considered as features in an untargeted fashion, which obviously is not the aim of this analysis. For instance: if I take the 0,5-1,0 MS2 scored metabolites into account of the Pos All tab in 428755_1_data_set_8048546_s09g5b.xlsx:

o MS2>0,5: Library; 51, Mona; 271, LipidMaps; 356. Total: 678 metabolites, not 8071.

3. Another reason for the many "identifications" is that the used LipidBlast database contains many isomers that are definitely not separated on the analytical system (isomers etc.). for example: HexCer-NDS d42:1 is found at two retention times where one is galactosylceramide and the other glucosylceramide with 10 min retention time difference. This is impossible as their retention time will be almost identical.

4. LPE 22:6 is annotated 5 times with retention times ranging from 14-34 min. Which one is correct (no MS2 score for either of them)? This is seen repeatedly in the dataset which further undermines my confidence in the data and the conclusions drawn from it.

5. I also have serious doubt about the ability of the lipidomics analysis to separate alkyl and alkenyl ether lipid species. Looking at the Pos_Compare_Total tab for example I see two features annotated as PC 38:5e, one as PC(O-18:1(9Z)/20:4(8Z,11Z,14Z,17Z)) and the other as PC(P-18:0/20:4(5Z,8Z,11Z,14Z)), thus the -O and -P isomers with the same exact mass. The retention time difference is 6 min. In our reversed phase analysis this difference is about 0,15 min. This annotation therefore cannot be correct.

As the O- vs. P- difference is an important finding in the lipidomics results which lead the authors to claim that siUBA1/siPEX influences this conversion. If annotation of these species is incorrect, which I consider likely for many of the annotations, the biological consequences inferred from them are too.

6. The authors state "A core set of 100 lipids and metabolites that are significantly and consistently modulated by both UBA1 and E2combo RNAi in a PEX-dependent manner included lipid species known to be regulated by peroxisomes". I could not find this list of lipids. These should be provided to allow evaluation of their peroxisomal nature.

7. In Pos_Compare_Total and Pos_All "Database name" and "LipidBlast name" do not always match (shifted?), this should be corrected.

I therefore strongly recommend to remove the metabolomics experiment as it has serious flaws that were not addressed in the revision. The conclusions drawn from them therefore, unfortunately, cannot be used as the data does not support it.

Response: We have followed the reviewer's suggestion and have removed the metabolomics experiment from the manuscript.

Reviewer #4 (Remarks to the Author):

The revised manuscript has addressed my comments. It is suitable for publication.

Response: Thank you for the positive evaluation of our work.